# Theoretically Understanding the Hidden Adversarial Price of Low-Rank Adaptation

## Abstract

Low-rank adaptation (LoRA) has emerged as a prominent parameter-efficient fine-tuning (PEFT) method for large pre-trained models, enabling strong downstream performance with minimal parameter updates. While LoRA is known to outperform head-only fine-tuning in terms of clean accuracy, its impact on adversarial robustness remains largely unexplored. In this work, and to the best of our knowledge, we present the first theoretical analysis of LoRA's adversarial robustness, comparing it to that of head-only fine-tuning. We formalize the notion of expected adversarial robustness and derive upper bounds demonstrating that, despite its superior clean performance, LoRA can be inherently less robust than head-only tuning due to the additional degrees of freedom introduced by its low-rank components. We further study the influence of LoRA's initialization scheme and show that simple changes in the initialization distribution of the low-rank matrix can significantly affect robustness. Finally, we support our theoretical findings with extensive experiments on both vision and language benchmarks under standard adversarial attacks. Our results provide a principled understanding of the trade-offs between parameter efficiency, clean performance, and adversarial robustness in commonly used fine-tuning strategies.

## 1 Introduction

Deep learning has led to significant breakthroughs across multiple domains, notably in computer vision (Dosovitskiy et al., 2021; Liu et al., 2021) and natural language processing (Devlin et al., 2019; Radford et al., 2019; Jiang et al., 2023), where foundation models have become central. These models, typically based on Transformer architectures (Vaswani et al., 2017), are pre-trained on large-scale datasets using auxiliary self-supervised tasks, enabling them to learn transferable representations. When fine-tuned, they achieve state-of-the-art performance on a wide range of downstream tasks. However, these foundation models are often extremely large, encompassing a lot of parameters that could range from millions to billions, which makes full fine-tuning both computationally expensive and impractical for many users. As a result, parameter-efficient fine-tuning (PEFT) (Han et al., 2024)

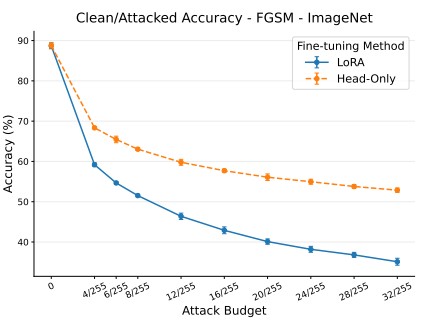

Figure 1: Clean/attacked accuracy on ImageNet subject to FGSM for a ViT.

strategies have gained attention. A common approach is to freeze the pretrained model and only train a lightweight classification or regression head (Kornblith et al., 2019; Chen et al., 2020). While efficient, this method often yields suboptimal downstream performance. To address this, Low-Rank Adaptation (LoRA) (Hu et al., 2022) has emerged as a leading PEFT technique. LoRA introduces learnable low-rank matrices into the model's weight structure, allowing it to adapt to downstream tasks while updating only a small subset of parameters. Empirically, LoRA often closely approaches the performance of full fine-tuning, making it a practical alternative for resource-constrained environments.

In parallel to advancements in finetuning methods, adversarial robustness remains a pressing challenge in deep learning. Neural networks are known to be vulnerable to small, carefully crafted

perturbations that can cause severe misclassifications, even when these perturbations are imperceptible to humans (Goodfellow et al., 2015). This vulnerability raises concerns in safety-critical applications such as autonomous vehicles, healthcare, and finance. While extensive research has been conducted on adversarial attack strategies (Tramer et al., 2020; Costa et al., 2024; Biggio et al., 2013) and defense mechanisms (Madry et al., 2017; Akhtar et al., 2021), the relationship between finetuning strategies and adversarial robustness remains underexplored. In particular, the majority of theoretical work has studied how LoRA's performance is influenced by hyperparameters such as rank (Kalajdzievski, 2023), learning rate (Hayou et al., 2024b), and initialization (Hayou et al., 2024a), while no theoretical work to date and to our knowledge has rigorously analyzed the impact of LoRA on adversarial robustness. Preliminary empirical evidence (such as observed in Figure 1) suggests LoRA may influence robustness, but a formal understanding of this phenomenon is lacking.

In this work, we aim to bridge the gap by investigating how LoRA-based fine-tuning affects adversarial robustness, specifically in comparison to fine-tuning using only a classification or regression head. While it is well established that LoRA outperforms head-only tuning in terms of clean accuracy, it remains unclear whether this gain comes at the cost of reduced robustness under adversarial attacks. To address this, we begin by formalizing the notion of expected adversarial robustness, which we then use to theoretically analyze and compare the robustness of head-only and LoRA-based fine-tuning. Our analysis leads to an upper bound suggesting that head-only fine-tuning exhibits stronger adversarial robustness than LoRA, primarily due to the additional parameters introduced by the low-rank adaptation layers. To further understand the influence of LoRA's design choices, we examine how its initialization scheme impacts robustness. In standard LoRA training, one of the low-rank matrices is initialized randomly while the other is set to zero; prior work (Hayou et al., 2024a) has shown that initializing $B$ to zero and $A$ randomly typically yields better clean accuracy. We extend this line of investigation by studying how varying the initialization of $A$ affects adversarial robustness and demonstrates that such a simple change can narrow the robustness gap between LoRA and head-only tuning. Finally, we empirically validate our theoretical findings on both vision and language benchmarks using standard adversarial attacks across multiple datasets. Our overall contributions can be summarized as follows:

- Using a formal notion of expected adversarial robustness, we theoretically show that head-only fine-tuning offers higher expected adversarial robustness than LoRA, due to the additional degrees of freedom introduced by LoRA's low-rank matrices.

- We analyze how LoRA's initialization scheme, particularly the initialization of its low-rank matrix $A$, and we consequently provide new additional insights on the choice of initial distribution, which could reduce the robustness gap between LoRA and head-only fine-tuning.

- We validate our theoretical findings through extensive experiments on vision and language tasks, using standard adversarial attacks and multiple benchmark datasets.

## 2 RELATED WORK

**Parameter-Efficient Fine-Tuning.** Most pre-trained models today are based on the Transformer architecture (Vaswani et al., 2017). Fully fine-tuning these large models for downstream tasks is often computationally expensive due to the sheer number of parameters, resulting in high memory and compute requirements. Parameter-Efficient Fine-Tuning (PEFT) aims to address this challenge by introducing a small number of trainable parameters, enabling efficient adaptation without updating the entire model. A simple approach is to fine-tune only the task-specific head, which reduces resource usage but often degrades performance. As an alternative, Low-Rank Adaptation (LoRA) (Hu et al., 2022), and its variants (Dettmers et al., 2023; Kopiczko et al., 2024; Hayou et al., 2024b; Li et al., 2024), inject a small set of trainable parameters into each layer of the frozen Transformer backbone, offering a better trade-off between parameter efficiency and downstream performance.

**Initialization of LoRA.** The initialization of the low-rank matrices in LoRA has recently received increased attention. Since the product of the two matrices is typically initialized to zero to preserve the behavior of the pre-trained model at the start of fine-tuning, various strategies have been proposed for initializing the non-zero matrix. Recent analysis (Hayou et al., 2024a) shows that this choice significantly influences optimization, with initializing $B$ to zero and $A$ randomly yielding

better average performance. AMT (Yang et al., 2024a) proposes an SVD-based initialization, aligning LoRA adapters with principal subspaces of the original weights to improve robustness under adversarial tuning. DoRA (Liu et al., 2024) further decomposes pre-trained weights into magnitude and direction, restricting LoRA updates to the directional component, leading to improved performance and stability compared to standard LoRA.

**Adversarial Robustness and LoRA.** Most prior work on LoRA has focused on its effectiveness for downstream task performance. However, recent studies have begun to explore the relationship between fine-tuning and adversarial robustness. In particular, works such as (Turbal et al., 2024; Yang et al., 2024b) empirically investigate the robustness of large language models in transfer settings. In the same direction, AutoLoRA (Xu et al., 2024) and ADV-LoRA (Wu et al., 2025) incorporate adversarial training, one of the most established techniques in robustness research, into the LoRA framework to enhance resilience. Despite these empirical advances, a theoretical understanding of how LoRA and its associated hyperparameters affect adversarial robustness remains lacking. This work aims to bridge that gap by developing a general theoretical framework linking LoRA to adversarial robustness, offering both theoretical and empirical insights that improve model resilience and open new research perspective.

## 3 PRELIMINARIES

In this section, we start by introducing some fundamental concepts that will be used afterwards in our work. Afterward, we formulate our problem setup, which will be considered in our analysis.

**Transformer-based Models.** Let $X \in \mathcal{X} \subseteq \mathbb{R}^{n \times d}$ denote a sequence of $n$ tokens, where each token $x_i \in \mathbb{R}^d$. The backbone of a Transformer $h : \mathcal{X} \subseteq \mathbb{R}^{n \times d} \to \mathcal{Z} \subseteq \mathbb{R}^{n \times d}$, as introduced in (Vaswani et al., 2017), is the *self-attention* mechanism, which computes a weighted combination of all token representations. Specifically, given learnable *query*, *key*, and *value* parameter matrices $W^Q, W^K, W^V \in \mathbb{R}^{d \times (d/H)}$, the output of a single *attention head* AH for input $X$ is defined as:

$$\mathrm{AH}(X) = \mathrm{softmax}\left( \frac{(XW^Q)(XW^K)^\top}{\sqrt{d/H}} \right) (XW^V), \tag{1}$$

where $H$ denotes the number of parallel attention heads and $d/H$ is the dimension per head. In practice, multiple attention heads $\mathrm{AH}_i$ are computed in parallel, then concatenated and projected using a learnable weight matrix $W^O \in \mathbb{R}^{d \times d}$, yielding the multi-head attention (MHA) operation:

$$\mathrm{MH}(X) = \mathrm{concat}\big(\mathrm{AH}_1(X), \mathrm{AH}_2(X), \dots, \mathrm{AH}_H(X)\big)W^O. \tag{2}$$

In addition, each Transformer block incorporates a residual connection (He et al., 2016), layer normalization (Ba et al., 2016) and a position-wise feed-forward network (FFN).

**Parameter-Efficient Fine-Tuning.** We focus on the fine-tuning stage, assuming a Transformer-based model pre-trained using any auxiliary task. For a downstream task, we are given labeled data $\mathcal{X} = (X_1, \dots, X_n)$ and corresponding labels $\mathcal{Y} = (y_1, \dots, y_n)$ to adapt the model. A simple approach is to train only a final classification or regression head while freezing the backbone, which is efficient but often suboptimal. Full fine-tuning of both encoder and head improves performance but requires substantial compute and memory. A recent alternative, Low-Rank Adaptation (LoRA) (Hu et al., 2022), introduces low-rank trainable matrices $A$ and $B$ while keeping the original weight matrix frozen. Specifically, for a dense layer weight $W \in \mathbb{R}^{d \times k}$, LoRA replaces it with:

$$W' = W + \frac{\alpha}{r} BA,$$

where $r$ is the rank, $\alpha$ a scaling factor, and $B \in \mathbb{R}^{d \times r}$, $A \in \mathbb{R}^{r \times k}$ are learned during fine-tuning.

**Problem Setup.** Without loss of generality, we consider a 1-layer Transformer-based model (TBM) where all activation functions are assumed to be 1-Lipschitz, which is the case for most commonly used activations. The input space is $\mathcal{X} \in [0, 1]^{n \times d}$, representing normalized data such as images. Fine-tuning is performed using an $L$-smooth loss function $\mathcal{L}$, optimized via gradient descent. Let $W_*$ denote the local optimum to which the model converges. For a learning rate $\eta \leq \frac{1}{L}$, the update rule for layer $i$ at step $t$ is:

$$W_{t+1}^{(i)} = W_t^{(i)} - \eta \nabla \mathcal{L}(W_t^{(i)}).$$

While we focus on gradient descent for clarity, the theoretical insights extend to other optimizers using similar analysis. Thus, our setup reflects a modeling choice rather than a limiting assumption.

# 4 ON THE ROBUSTNESS OF LoRa

In this section, we aim to theoretically understand the connection between LoRA finetuning and the resulting adversarial robustness, taking the head-only finetuning as a basis for comparison. We start by formalizing the concept of expected adversarial robustness and, consequently, derive theoretical insights for both the head-only finetuning and the LoRA counterpart, showcasing the difference in terms of adversarial robustness.

## 4.1 ADVERSARIAL ROBUSTNESS

In this work, we focus on evasion attacks (Biggio et al., 2013; Pitropakis et al., 2019), which consist of attacking the model at test or inference time. We consider that this setting is more adapted to real-world scenarios, where in the majority of cases, the final user/attacker only has access to the model at inference time. In this direction, let's consider a trained classifier $f : \mathcal{X} \to \mathcal{Y}$ and let $x \in \mathcal{X}$ be an input with its associated label vectors $y \in \mathcal{Y}$, such that $f(x) = y$. The goal of an attacker is to craft a small additional perturbation to the input, such as to generate a point $\tilde{x}$ whose prediction $f(\tilde{x})$ is different from the original one. We note that the generated adversarial perturbation should be similar to the original input, and therefore, we need to consider a similarity budget $\epsilon$, together with the corresponding distance. For our current study, we consider the $\ell_2$ distance and consequently define our attack neighborhood of our input $x$ with respect to an attack budget $\epsilon$ as:

$$\mathcal{B}(x, \epsilon) = \{\tilde{x} \in \mathcal{X} : \|x - \tilde{x}\| \leq \epsilon\}$$

Given the previous neighborhood, the attacker aims to find within that neighborhood the points that not only satisfy the adversarial aim of flipping the classification but also result in the worst prediction. In this direction, given a finetuning strategy $\zeta$ which is applied to our considered pre-trained model $f$, the *adversarial risk* can be formulated as follows:

$$\mathcal{R}_\epsilon[f, \zeta] = \mathbb{E}_{x \in \mathcal{D}_\mathcal{X}} \left[ \sup_{\tilde{x} \in \mathcal{B}(x, \epsilon)} d_\mathcal{Y} \left( \zeta_f(\tilde{x}), \zeta_f(x) \right) \right]. \tag{3}$$

with $d_\mathcal{Y}$ being any defined distances in the measurable output $\mathcal{Y}$. In the current work, and similar to the input space, we consider $\ell_2$-norm as our distance metric for the output space. Note that there exists an equivalence in terms of norm, and therefore, this latter choice can easily be extended to other norms and doesn't limit our provided insights in any direction.

From an adversarial defense perspective, the objective is to ensure that the previously introduced risk remains small, implying that it's harder to find a perturbation within the considered budget $\epsilon$, and consequently that the model predictions are stable within that neighborhood, reflecting the adversarial robustness of the model. We can formalize this notion for a finetuning strategy as follows:

**Definition 1** (Adversarial Robustness). *The finetuning strategy $\zeta$ is said to be $(\epsilon, \gamma)$-robust if its adversarial risk with respect to the classifier $f$ satisfies: $\mathcal{R}_\epsilon[f, \zeta] \leq \gamma$.*

We note that we approach the theoretical analysis from an upper-bound perspective (denoted as $\gamma$), since it is hard to compute the exact adversarial risk value. Obviously, the smaller the upper-bound, the more robust the model is expected to be, and therefore by comparing the two quantities, we can have an idea about the performance of the two considered finetuning strategies.

## 4.2 ON THE ROBUSTNESS OF LOW-RANK ADAPTATION

Building on the formal framework introduced previously, we now analyze the adversarial robustness of Low-Rank Adaptation (LoRA) in comparison to the standard head-only finetuning strategy. While LoRA is widely recognized for its effectiveness in improving downstream performance, often measured by clean accuracy, this gain comes from its ability to modify a larger subset of the model's parameters, including those within internal Transformer components. In contrast, head-only finetuning restricts adaptation to the final classification layer, preserving the backbone of the pre-trained

model. This difference in parameter access raises, consequently, a natural question: Does the increased expressivity provided by LoRA come at the cost of adversarial robustness? Typically, while LoRA allows for better task-specific adaptation, it may also expose the model to increased vulnerability under test-time perturbations. To study this trade-off systematically, we adopt the notion of expected adversarial risk defined earlier, and derive upper bounds for both finetuning strategies under the same theoretical problem setup. Specifically, we consider a pre-trained Transformer-based model (TBM) denoted by $f$, and analyze its behavior under both head-only and LoRA-based finetuning. We consider $f$ as a one-layer Transformer block with $H$ dot-product self-attention heads, following the structure and the notations outlined in Section 3.

**Proposition 1.** *Let $f \colon \mathcal{X} \to \mathcal{Y}$ be a pre-trained TBM-based model following the problem setup. The head-only finetuning strategy $\zeta_f^{Head\text{-}only}$ is $(\epsilon, \gamma)$-robust, with: $\gamma_{Head\text{-}Only} = \left(\frac{d}{d-1}\right)^2 C_1 C_2 \epsilon$,*

$$C_1 = 1 + \|W_O\|\sqrt{H}\max_h \left[\|W^{V,h}\|\left[\tfrac{4}{\sqrt{d/H}}\|W^{Q,h}\|\|W^{K,h}\|+1\right]\right],\ C_2 = \left(1 + \|W_{FFN}\|\right)\|W_{out}\|.$$

Proposition 1 provides a concrete expression for the adversarial risk bound under head-only finetuning, which depends explicitly on the norms of the model's attention and feedforward weights. In the following, we extend this analysis to the case of LoRA-based finetuning, applying the same theoretical approach in order to establish a basis for comparison between the two strategies. Specifically, we consider that the LoRA is only applied to the query (Q) and value (V) projection matrices of the attention mechanism, as in the original proposed work.

**Theorem 1.** *Let $f \colon \mathcal{X} \to \mathcal{Y}$ be the pre-trained TBM-based model following the considered problem setup. For the LoRA-based finetuning strategy $\zeta_f^{LoRA}$, where the LoRA is only applied to the main Transformer part, is $(\epsilon, \gamma)$-robust, with: $\gamma_{LoRA} = \left(\frac{d}{d-1}\right)^2 C_1' C_2 \epsilon$, where:*

$$C_1' = 1 + \|W_O\|\sqrt{H}\max_h \left(\left[\|W^{V,h}\| + \tfrac{\alpha}{r}\|A^{V,h}\|\|B^{V,h}\|\right]\left[\tfrac{4}{\sqrt{d/H}}\left(\|W^{Q,h}\| + \tfrac{\alpha}{r}\|A^{Q,h}\|\|B^{Q,h}\|\right)\|W^{K,h}\| + 1\right]\right).$$

The derived upper bounds in Proposition 1 and Theorem 1 provide a comparative theoretical framework for evaluating the adversarial robustness of head-only finetuning versus LoRA. While both bounds scale linearly with the perturbation radius $\epsilon$ and share a similar structural dependence on network norms, the LoRA bound introduces additional terms involving the norms of the low-rank adaptation matrices $A$ and $B$, scaled by the factor $\alpha/r$. These terms effectively inflate the expected adversarial risk of the model when subject to input perturbations, yielding a looser (i. e., higher) upper bound on the adversarial risk $\mathcal{R}_\epsilon[f, \zeta]$ under LoRA.

This difference is intuitive and arises from the core design of LoRA, which introduces learnable low-rank updates into the internal weight matrices, specifically within the query, key, and value projections of the self-attention mechanism. By modifying these internal components, LoRA increases the space of adaptable parameters, enhancing task-specific expressivity and improving clean accuracy. However, this also creates additional pathways through which input perturbations can affect the output, making the model more vulnerable to adversarial attacks. In contrast, head-only finetuning restricts adaptation to the final classification layer, leaving the backbone unchanged and preserving the stability of the pre-trained representations. We consider that these results can also be categorized on the general robustness-performance trade-off, where, by aiming to have better performance, the model's boundaries are adapted to the task, resulting in richer task-specific adaptation, but at the cost of amplifying the model's response to small input perturbations. From a theoretical standpoint, this trade-off is captured directly by the looser robustness bound. Practically, it suggests that while LoRA may be preferable when downstream accuracy is the sole objective, it may lead to weaker performance in adversarial settings where robustness is critical. Specifically, for a final user, before choosing the right finetuning approach, an analysis of the objectives and the trade-off between clean and attacked accuracy should be done.

**Extension to Multi-Layer Transformers.** Although our theoretical analysis focuses on a single-layer Transformer-based model $f$, the results naturally extend to the multi-layer case. Specifically, a Transformer model with $L$ layers, denoted as $f^{(L)}$, can be expressed as a composition of $L$ single-layer functions: $f^{(L)}(x) = f^{(L-1)} \circ f^{(L-2)} \circ \cdots \circ f^{(1)}(x)$. Under this formulation, and following

standard results from Lipschitz continuity, the overall adversarial risk bound $\gamma$ for either finetuning strategy becomes a multiplicative composition of the bounds for each individual layer. That is, the robustness bound compounds across layers, maintaining the same structural form as in the single-layer case. We additionally note that the underlying assumptions of our problem setup (Section 3) still hold in the multi-layer setting. Since each layer operates on bounded activations (as we assume $\mathcal{X} \subseteq [0, 1]^{n \times d}$), the composition of bounded functions preserves theoretical soundness. As a result, our robustness framework remains applicable in deeper architectures.

## 5 CONNECTING INITIALIZATION TO LORA'S ROBUSTNESS

In the previous section, we theoretically established a notable gap in adversarial robustness between head-only and LoRA-based finetuning strategies. Specifically, our analysis showed that LoRA exhibits a higher expected adversarial risk, suggesting reduced robustness to test-time perturbations. Motivated by this observation, we now turn to investigating whether this robustness gap can be influenced, or potentially mitigated and reduced, by specific choices in LoRA's hyperparameters. In particular, we focus on how the *initialization of the low-rank matrices $A$ and $B$, which define the LoRA updates, impacts adversarial robustness.*

The effect of initialization (Hayou et al., 2024a) in LoRA has recently received increased attention. By design, the product of the two matrices is intended to start at zero to ensure that the training starts from the original weights of the pre-trained model. However, recent empirical studies (Hayou et al., 2024a) suggest that initializing $A$ with random values and setting $B$ to zero tends to yield better generalization and downstream performance than the reverse configuration. While these findings pertain to clean accuracy, their implications for adversarial robustness remain underexplored. In this section, we extend this line of inquiry by studying how the randomness in the initialization distribution of matrix $A$, which governs the initial adaptation direction, affects the final adversarial robustness of the finetuned model. Our goal is to understand whether certain initialization choices introduce more sensitivity to adversarial perturbations, and whether controlling the variance or structure of this randomness can lead to more robust LoRA configurations.

In this perspective, we consider the same setting as the one studied in the previous section, where $f$ is a 1-Layer Transformer-based model and the aim is to link the initial weights with the resulting upper-bound on the expected adversarial robustness.

**Theorem 2.** *Let $f \colon \mathcal{X} \to \mathcal{Y}$ be our pre-trained TBM-based model. Let's consider the LoRA finetuning strategy, where all the low-rank matrices $A$ in layer $h$ are initialized as $A_0^{Q,h}$ (for Query) and $A_0^{V,h}$ (for Values), then the resulting $C_1'$ constant in $\gamma_{LoRA}$ ( Theorem 1) can be written as:*

$$C_1' \leq K_1 \left(1 + \eta L\right)^t \max_h \|A_0^{(V,h)}\| + K_2 (1 + \eta L)^{2t} \max_h \|A_0^{(V,h)}\| \|A_0^{(Q,h)}\| + C,$$

*with $K_1, K_2$ and $C$ being constants depending on the final weight norms (derived in Equation 12).*

We observe that the upper bound derived in Theorem 2 directly links the norm of the chosen initialization matrix to the constant $C_1'$, which in turn influences $\gamma_{LoRA}$ and thereby the model's adversarial robustness. This result highlights that initialization, often treated as a secondary detail, plays a critical role in shaping LoRA's robustness characteristics and should be carefully designed. Since the initialization also affects the model's downstream performance, finding an appropriate trade-off between robustness and clean accuracy becomes crucial. In particular, the initialization of $A$ should be designed to balance these objectives, enabling the construction of LoRA-based models that are both performant and robust. To better showcase the practical aspect of our theoretical result, we consider a practical application where we consider that the matrices are initialized from a Uniform distribution $\mathcal{U}(-a, a)$, where $a$ is a parameter.

**Lemma 1.** *Consider LoRA matrices, with rank $r$, and for each head $h = 1, \ldots, H$ initialized with entries drawn i.i.d. from $\mathcal{U}(-a, a)$ independently across heads and stacks. Then the expected value of the robustness constant $C_1'$, derived in Theorem 1, satisfies:*

$$\mathbb{E}[C_1'] = \mathcal{O}\left( (1 + \eta L)^{2t} a^2 \left( (\sqrt{r} + \sqrt{k}) + \sqrt{\log H} \right)^2 \right).$$

The result of Lemma 1 establishes a direct relationship between the initialization parameter $a$ of the Uniform distribution and the resulting upper bound on the expected adversarial robustness. Although

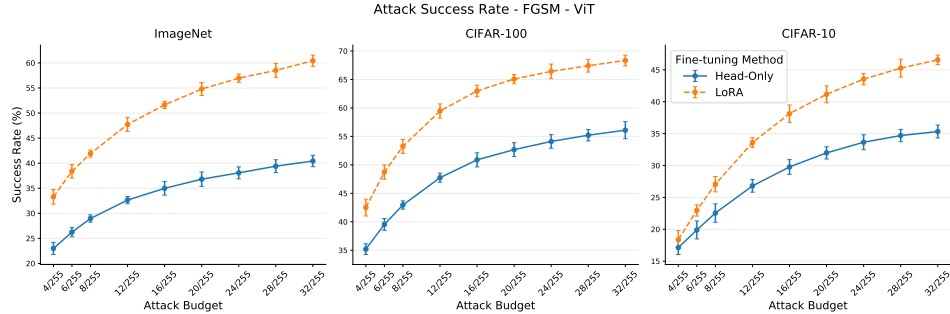

Figure 2: Success Rate of FGSM Attack on a ViT for different datasets and attack budgets.

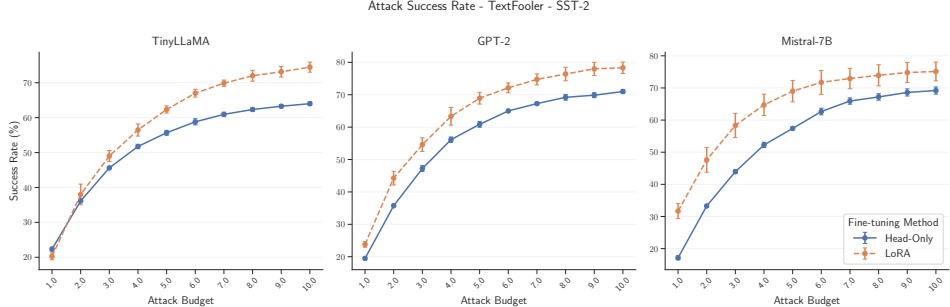

Figure 3: Attack Success Rate of the TextFooler Attack on TinyLLaMA, GPT-2, and Mistral-7B when applied to SST-2 dataset with different attack budget (number of words changed).

the analysis focuses on the Uniform case, a similar adaptation of Theorem 2, and similar reasoning can be extended to other initialization distributions.

## 6 EMPIRICAL EVALUATION

We empirically validate our theoretical insights using standard adversarial attacks across two widely used modalities: images and text. We start by outlining the experimental setup for both domains.

**Computer-Vision.** We have chosen to operate under two mainly widely used models, namely the Vision Transformer (ViT) (Dosovitskiy et al., 2021), which was the basis of our theoretical study, and the SwiN Transformer (Liu et al., 2021). For both models, we have considered the two mainly used adversarial attacks in this domain, which are the Fast Gradient Sign Method (FGSM) and the Proximal Gradient Descent (PGD), focusing on image classification using the CIFAR-10, CIFAR-100 (Krizhevsky et al., 2009), and ImageNet-100 (Russakovsky et al., 2015).

**Natural Language Processing (NLP).** We have chosen to operate through a number of different models, namely Bert-Base (Devlin et al., 2019), DistilBert-Base (Sanh et al., 2019), GPT2 (Radford et al., 2019), Gemma-2B (Team et al., 2024), llama3_2_1B (Dubey et al., 2024), Tiny-Llama (Zhang et al., 2024), and Mistral-7B (Jiang et al., 2023). For all models, we perform TextFooler (Jin et al., 2020) attack, and for some models, we also perform A2T (Yoo & Qi, 2021) attack. We focus the evaluation on the text classification task using IMDb (Maas et al., 2011), SST-2 (Socher et al., 2013), and Yelp Polarity (Zhang et al., 2015) datasets.

**Considered Metrics.** For both modalities, we report the clean/attacked accuracy and the success rate, which is the number of samples that were successfully attacked, meaning that the attack was successful in finding a perturbation within the budget that was able to flip the original classification.

We note that for all the models, the LoRA adaptation is applied to the whole self-attention component. The code to reproduce our results and experiments is provided in the Supplementary Materials, and additional details about the hyperparameters and the problem setup are provided in Appendix E.

## 6.1 EXPERIMENTAL RESULTS

**Image-Based Evaluation.** Figure 2 (respectively Figure 8 in Appendix D.1) presents the average success rate, and the corresponding standard deviations, of the FGSM attack for both head-only and LoRA finetuning strategies on a ViT (and SwiN, respectively), evaluated across multiple datasets and perturbation budgets. The empirical results align with our theoretical findings: across all datasets, LoRA consistently yields a higher attack success rate, indicating lower adversarial robustness compared to head-only finetuning. Notably, the robustness gap between the two strategies can be substantial. For example, on ImageNet, the difference in attack success rate can reach up to 20%, despite a clean accuracy gap of only around 1%. This performance contrast highlights that even small gains in clean performance under LoRA may come at a significant cost in adversarial settings. Similar insights are observed for the CIFAR dataset family, where by aiming for a small increase in clean robustness (around $3 - 4\%$), the resulting success rate can reach around $12 - 15\%$.

Table 1: Average Clean Accuracy and Success Rate ($\pm$ standard deviation) of the ViT and SwiN for both head-only and LoRA finetuning subject to FGSM and PGD attack on different datasets.

| Model | Dataset | Strategy | Clean Accuracy ↑ | Success Rate (FGSM) ↓ | Success Rate (PGD) ↓ |
|---|---|---|---|---|---|
| ViT | ImageNet | Head-Only | $88.7 \pm 0.2$ | $28.9 \pm 0.7$ | $84.2 \pm 0.4$ |
| | | LoRA | $88.9 \pm 0.1$ | $41.9 \pm 0.8$ | $96.4 \pm 0.3$ |
| | CIFAR-10 | Head-Only | $97.4 \pm 0.1$ | $22.5 \pm 0.3$ | $88.4 \pm 0.6$ |
| | | LoRA | $98.6 \pm 0.1$ | $27.1 \pm 0.6$ | $93.1 \pm 0.3$ |
| | CIFAR-100 | Head-Only | $87.9 \pm 0.9$ | $42.9 \pm 0.8$ | $92.4 \pm 0.7$ |
| | | LoRA | $90.8 \pm 0.2$ | $53.2 \pm 0.4$ | $96.2 \pm 0.8$ |
| SwiN | ImageNet | Head-Only | $89.8 \pm 0.2$ | $29.9 \pm 0.8$ | $90.2 \pm 0.4$ |
| | | LoRA | $90.3 \pm 0.1$ | $36.6 \pm 0.6$ | $94.8 \pm 0.7$ |
| | CIFAR-10 | Head-Only | $97.8 \pm 0.1$ | $26.1 \pm 0.8$ | $93.4 \pm 0.2$ |
| | | LoRA | $98.5 \pm 0.1$ | $28.9 \pm 0.7$ | $95.1 \pm 0.4$ |
| | CIFAR-100 | Head-Only | $87.6 \pm 0.1$ | $45.3 \pm 0.6$ | $94.2 \pm 0.3$ |
| | | LoRA | $92.1 \pm 0.3$ | $50.2 \pm 0.4$ | $97.4 \pm 0.2$ |

**Text-Based Evaluation.** Figure 3 (and Figure 13 - Appendix D.3) presents the average success rates, along with standard deviations, for the TextFooler and A2T adversarial attacks applied to models fine-tuned using either head-only or LoRA strategies across varying perturbation budgets. The results in the NLP setting closely mirror the trends observed in computer vision, further reinforcing the generality of our theoretical findings across modalities. In addition, Table 2 summarizes both clean accuracy and attack success rates under a fixed perturbation budget of 3 word substitutions. Across all evaluated models, LoRA fine-tuning generally shows lower robustness compared to head-only tuning. Additional experiments on other architectures and models are provided in Appendix D.3.

Table 2: Average Clean Accuracy and Success Rate ($\pm$ standard deviation) of BERT, DistilBERT and GPT-2 for head-only and LoRA finetuning subject to TextFooler and A2T on different datasets.

| Model | Dataset | Strategy | Clean Accuracy ↑ | Success Rate (TextFooler) ↓ | Success Rate (A2T) ↓ |
|---|---|---|---|---|---|
| BERT | IMDb | Head-Only | $83.2 \pm 0.9$ | $11.3 \pm 0.1$ | $8.6 \pm 0.6$ |
| | | LoRA | $90.5 \pm 0.9$ | $16.7 \pm 1.0$ | $14.9 \pm 0.1$ |
| | SST-2 | Head-Only | $83.4 \pm 0.5$ | $54.0 \pm 0.0$ | $28.1 \pm 0.7$ |
| | | LoRA | $92.1 \pm 0.2$ | $53.4 \pm 1.6$ | $22.9 \pm 0.6$ |
| | Yelp Polarity | Head-Only | $86.1 \pm 1.5$ | $14.7 \pm 2.2$ | $11.0 \pm 1.1$ |
| | | LoRA | $92.6 \pm 0.5$ | $15.9 \pm 0.8$ | $11.5 \pm 0.7$ |
| GPT-2 | IMDb | Head-Only | $85.7 \pm 1.0$ | $6.7 \pm 0.8$ | $11.0 \pm 2.0$ |
| | | LoRA | $91.9 \pm 0.9$ | $7.9 \pm 0.4$ | $13.4 \pm 2.3$ |
| | SST-2 | Head-Only | $82.1 \pm 0.5$ | $47.2 \pm 0.9$ | $30.1 \pm 0.4$ |
| | | LoRA | $91.0 \pm 1.3$ | $54.6 \pm 2.1$ | $22.0 \pm 0.8$ |
| | Yelp Polarity | Head-Only | $85.2 \pm 1.6$ | $10.1 \pm 0.8$ | $12.3 \pm 2.3$ |
| | | LoRA | $92.4 \pm 0.5$ | $7.9 \pm 0.8$ | $8.9 \pm 1.4$ |

These findings underscore the practical significance of our theoretical analysis. While LoRA improves downstream performance in terms of clean accuracy, it also introduces increased vulnerability to adversarial perturbations. A key insight is that the gains in clean accuracy offered by LoRA

come at an adversarial cost, with performance degrading more severely under attack. This observation highlights an important trade-off between clean accuracy and robustness, particularly in safety-critical applications where reliability under distribution shift or adversarial threat is paramount. In such contexts, clean accuracy alone is an insufficient metric and must be complemented by robustness evaluations.

## 6.2 Effect of Hyper-Parameters

### 6.2.1 Effect Of Initialization

We further investigate the impact of initialization strategies to demonstrate the practical relevance of the theoretical insights from Section 5, particularly Theorem 2. To this end, we evaluate several initialization schemes for the LoRA matrix $A$. Specifically, we consider the default *Kaiming* initialization used in the PEFT package, the well-known *Xavier* initialization, and three additional classical distributions: *Gaussian*, *Orthogonal*, and *Uniform*.

Figure 4 reports the average clean and attacked accuracies across various adversarial budgets and initialization distributions. As anticipated, the choice of initialization significantly influences the final adversarial robustness. Although all distributions yield similar clean accuracies (within a $2\%$ range), the attacked accuracies show a gap of up to $10\%$ between the most and least robust initializations (Uniform versus Kaiming). These results indicate that selecting an appropriate initialization can substantially

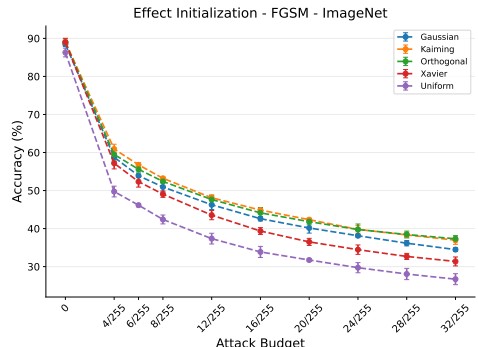

Figure 4: Effect of the chosen initialization distribution on the resulting Attacked Accuracy of ImageNet.

enhance robustness without sacrificing clean performance, thereby reducing the robustness gap between the LoRA and head-only finetuning strategies. Note that the additional results for the other datasets are provided in Figure 11 (Appendix D.1). We further examine how the choice of initial weight norm influences model vulnerability and adversarial robustness. Figure 5 reports results obtained by varying the scaling factors of both Kaiming and Orthogonal initialization schemes. For Kaiming initialization, note that the scaling factor is inversely related to the resulting weight norm (via $\sqrt{2/(1+a^2)}$). Consistent with our theory, increasing the scaling parameter leads to larger initial weight norms, which in turn produce higher $\gamma$ values and reduced adversarial robustness.

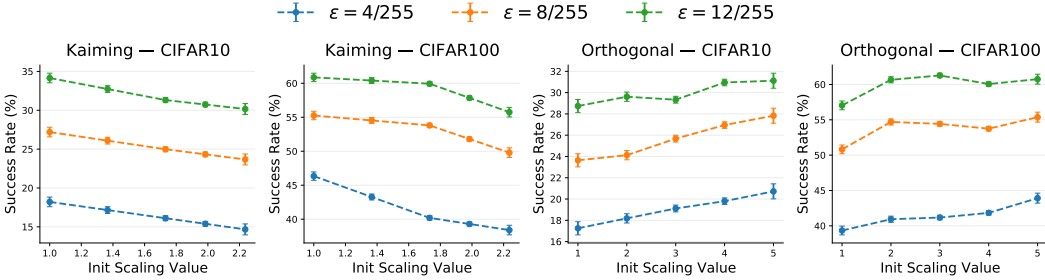

Figure 5: Effect of varying the initial weight norm, controlled through the scaling parameter, on the FGSM success rates for CIFAR-10 and CIFAR-100.

### 6.2.2 Effect of LoRA Scaling

LoRA fine-tuning depends on two key hyperparameters: the rank $r$ of the learnable matrices $A$ and $B$, and the scaling factor $\alpha$. As shown in Theorem 1, the robustness bound $\gamma_{\text{LoRA}}$ is directly affected by these parameters. To validate this relationship empirically, we fix the rank to $r = 4$ and vary $\alpha$ to assess its impact on adversarial vulnerability.

Figure 6 presents the average attack success rates (with standard deviation) on CIFAR-10 and CIFAR-100 for different values of $\alpha$. Consistent with the theoretical insights that larger values of $\alpha$ increase the upper bound, the empirical results confirm that increasing $\alpha$ leads to higher attack success rates and thus reduced robustness. Interestingly, the widely adopted practice of setting $\alpha = r$ appears suboptimal. Instead, using a smaller value such as $\alpha = 1$ yields a reduction of approximately $10\%$ in success rate, enhancing robustness and narrowing the gap with head-only fine-tuning.

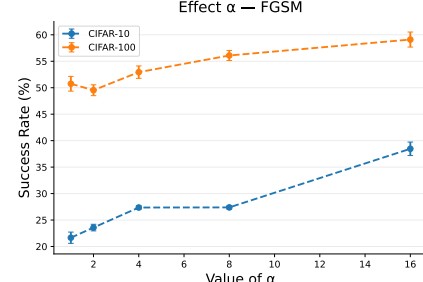

Figure 6: Effect of the LoRA parameter $\alpha$ on the resulting attack success rate.

### 6.3 GENERALIZING TO OTHER LoRA ADAPTATIONS

Beyond the original LoRA formulation introduced by Hu et al. (2022), several extensions have been proposed to advance parameter-efficient fine-tuning. Our theoretical analysis focuses on this canonical variant to isolate how its parameterization influences adversarial robustness, but the resulting upper bound also suggests that certain adaptations may confer additional robustness benefits. For instance, DeLoRA Bini et al. (2025) and NB-LoRA

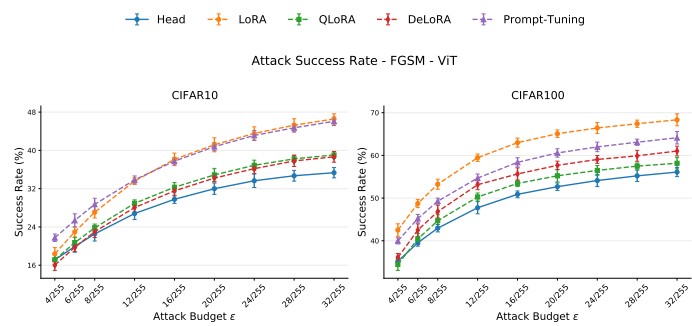

Figure 7: Effect of the LoRA parameter $\alpha$ on the resulting attack success rate.

Wang et al. (2025) explicitly constrain the norm of the learned updates, which, when combined with our bound, implies a secondary robustness effect arising from tighter control of update magnitudes. Likewise, QLoRA Dettmers et al. (2023) employs 4-bit quantization to reduce memory usage while preserving downstream performance, and such quantization may incidentally lead to improved robustness. To evaluate these hypotheses, we empirically compare the adversarial robustness of standard LoRA with several of these adaptations. Finally, to illustrate how other PEFT strategies behave under adversarial perturbations, we also examine Prompt Tuning, in which a small set of learned continuous embeddings added to the input rather than modifying model weights, serves as the only trainable component. Figure 7 provides the empirical findings. As expected from our theoretical study, norm-bounded variants achieve higher robustness than standard LoRA, yet they still fall short of the robustness achieved by head-only fine-tuning. Additional results for NLP are provided in Figure 14.

## 7 CONCLUSION

In this work, we present the first theoretical analysis that explores the connection between LoRA as a fine-tuning strategy and the adversarial robustness of the resulting model. Our theoretical findings, supported by empirical results, indicate that the gains in clean accuracy achieved through LoRA come at the cost of increased vulnerability to adversarial attacks, particularly when compared to head-only fine-tuning. However, our analysis also highlights the important role of hyperparameters, specifically the scaling factor $\alpha$ and the initialization scheme, in shaping this trade-off. We show that appropriate choices of these parameters can significantly reduce the robustness gap, yielding a more favorable balance between clean and attacked accuracy *without introducing additional constraints or computational overhead*, effectively offering a "free-lunch" improvement.

**Limitations.** While our work focuses on offering theoretical guidance for tuning LoRA's hyperparameters, we believe it opens a new direction for designing LoRA variants that are not only effective on downstream tasks but also inherently more robust to adversarial perturbations.

## ETHICS STATEMENT

In this paper we study the adversarial robustness of LoRA-based models using only openly available datasets and pretrained models. Our work does not involve human subjects and therefore does not require IRB approval. All datasets used are publicly available and appropriately licensed. Although adversarial attacks are employed, they are standard, publicly available methods used solely to evaluate and improve model robustness. We believe that examining how these models respond to adversarial inputs is an important part of responsible AI research. By highlighting potential weaknesses, this line of work can help the community build systems that are more reliable, secure, and less vulnerable to misuse. In this context, to the best of our knowledge, this research does not raise ethical concerns related to discrimination, bias, privacy, or security. No conflicts of interest or legal compliance issues are associated with this work. We additionally note that LLMs were used only to assist with text refinement.

## REPRODUCIBILITY STATEMENT

We have made an effort to ensure that our results can be reproduced by others. All datasets and pretrained models we use are publicly available and are clearly referenced in the paper. The experimental setup, including how LoRA models are fine-tuned and how adversarial evaluations are carried out, is described in detail in the main text and the appendix (mainly Appendix E). The considered theoretical problem setup is clearly explained in Section 3 and all the theorem's proofs and extended results are included in the appendix. Finally, to support independent verification, the code to reproduce our results is included in the Supplementary Materials and shall be made public upon publication.

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

# Supplementary Material

## A    PROOF OF PROPOSITION 1

**Proposition.** *Let $f \colon \mathcal{X} \to \mathcal{Y}$ be the pre-trained TBM-based model following the considered problem setup. The head-only finetuning strategy $\zeta_f^{\text{Head-only}}$ is $(\epsilon, \gamma_{\text{Head-Only}})$-robust, with:*

$$\gamma_{\text{Head-Only}} = \left( \frac{d}{d-1} \right)^2 C_1 C_2 \epsilon,$$

$$\text{with: } C_1 = \left( 1 + \|W_O\| \sqrt{H} \max_h \left[ \|W^{V,h}\| \left[ \frac{4}{\sqrt{d/H}} \|W^{Q,h}\| \|W^{K,h}\| + 1 \right] \right] \right),$$

$$C_2 = \left( 1 + \|W_{FFN}\| \right) \|W_{out}\|,$$

*and $W$ being the different weights of the models (as explained in Section 3).*

*Proof.* Let's consider our input $X \in \mathcal{X}$ composed of $n$ tokens $x_i \in \mathbb{R}^d$. We consider that our model $f$ is built using the dot-product self-attention as referred to in Equation 1 and reformulated as:

$$\text{AH}(x) = \text{Softmax}\left( \frac{(XW^Q)(XW^K)^T}{\sqrt{\frac{D}{H}}} \right) XW^V$$

$$= PXW^V = h(X)W^V,$$

where $W^Q, W^K, W^V$ are learnable weights of the model. Let's consider the function $h(X)$, we can write:

$$f(X) = PX = \text{Softmax}(XA^T X^T)X$$

$$f(X) = PX = \text{Softmax}\left( XA^\top X^\top \right) X = \begin{bmatrix} h_1(X)^\top \\ \vdots \\ h_n(X)^\top \end{bmatrix} \in \mathbb{R}^{n \times d}, \quad \text{with:}$$

$$A = \frac{W^K W^{Q^\top}}{\sqrt{d/H}} \in \mathbb{R}^{d \times d} \quad \text{and} \quad h_i(X) = \sum_{j=1}^{n} P_{ij} x_j \quad \text{with} \quad P_i^\top = \text{Softmax}(XAx_i).$$

By analyzing the partial derivatives, we can directly write the following regarding eh Jacobian matrix of $h$:

$$J_{ij} = X^\top P^{(i)} E_{ji} X A^\top + \delta_{ij} \left( X^\top P^{(i)} X A \right) + P_{ij} I_d,$$

with:

- $P^{(i)} = \text{diag}\left( P_{i:} \right) - P_{i:}^\top P_{i:}$, [Softmax derivate]

- $E_{ji}$ is the $(n \times n)$ matrix with a single 1 in position $(j, i)$.

Based on this, two elements arises:

$$\text{If } i \neq j, \quad J_{ij} = X^\top P^{(i)} E_{ji} X A^\top + P_{ij} I, \tag{4}$$

$$\text{If } i = j, \quad J_{ii} = X^\top P^{(i)} E_{ii} X A^\top + X^\top P^{(i)}, X, A + P_{ii} I. \tag{5}$$

We recall that the input images are considered to be normalized, and therefore we can write:

$$\|X\| \leq 1$$

Additionally, since $P_{i:}$ is the output of the softmax, then can be considered a probability distribution. Therefore, $\sigma_{max}(diag(p)) \leq 1$ and $pp^T$ has rank 1:

$$\|P^{(i)}\| = \|\text{diag}(P_{i:}) - P_{i:}^\top P_{i:}\| \leq 2$$

**Case 1.** We start by considering the first case $i \neq j$, in which we have:

$$J_{ij} = X^\top P^{(i)} E_{ji} X A^\top + P_{ij} I.$$

Consequently we have the following:

$$\begin{aligned}
\|J_{ij}\| &\leq \|X^\top P^{(i)} E_{ji} X A^\top\| + \|P_{ij} I\| \\
&\leq 2 \times \|A\| + 1 \\
&\leq \|A\| + 1
\end{aligned}$$

**Case 2.** For the second case $i = j$, we have the following:

$$J_{ii} = X^\top P^{(i)} E_{ii} X A^\top + X^\top P^{(i)} X A + P_{ii} I.$$

We apply the same analogy as the previous case:

$$\begin{aligned}
\|J_{ii}\| &\leq \|X^\top P^{(i)} E_{ii} X A^\top\| + \|X^\top P^{(i)} X A\| + \|P_{ii} I\| \\
&\leq 2\|A\| + 2\|A\| + 1 \\
&\leq 4\|A\| + 1
\end{aligned}$$

So overall, we have the following:

$$\|J_{ij}\|_{op} \leq \begin{cases} 2\|A\| + 1, & \text{if } i \neq j, \\ 4\|A\| + 1, & \text{if } i = j. \end{cases}$$

So with our theoretical assumptions, the Jacobian is bounded and we have: $\mathcal{L}_h \leq 4\|A\| + 1$.

Specifically, for an attention head $h$, we have the following computation taking into account the different learnable weights:

$$\mathcal{L}_{head} \leq \|W^{V,h}\| \Big[ \frac{4}{\sqrt{d/H}} \|W^{Q,h}\| \|W^{K,h}\| + 1 \Big]$$

Since $f$ is represented by $H$ separate attention head, then their concatenated output as explained in Equation 2 is subject to the following:

$$\begin{aligned}
\mathcal{L}_{MH} &\leq \|W_O\| \sqrt{H} \max_h \big[ \mathcal{L}_{head} \big] \\
&\leq \|W_O\| \sqrt{H} \max_h \Big[ \|W^{V,h}\| \Big[ \frac{4}{\sqrt{d/H}} \|W^{Q,h}\| \|W^{K,h}\| + 1 \Big] \Big]
\end{aligned}$$

Finally, we need to take into account the application of the FFN and LN (with its parameters $\gamma = 1$ and $\beta = 1$). In addition, giving that we consider the head-only fine-tuning strategy, we consider that a final linear layer $W_{out}$ is trained. Since ReLU is 1-Lipschitz, we have the following result:

$$\begin{aligned}
\mathcal{L}_f &\leq L_{LN}^2 (1 + \mathcal{L}_{MH})(1 + L_{FFN}) \\
&\leq \Big( \frac{d}{d-1} \Big)^2 \big( 1 + \mathcal{L}_{MH} \big) \big( 1 + \|W_{FFN}\| \big) \\
&\leq \Big( \frac{d}{d-1} \Big)^2 \Big( 1 + \|W_O\| \sqrt{H} \max_h \big[ \|W^{V,h}\| \big[ \frac{4}{\sqrt{d/H}} \|W^{Q,h}\| \|W^{K,h}\| + 1 \big] \big] \Big) \big( 1 + \|W_{FFN}\| \big) \\
&\leq \Big( \frac{d}{d-1} \Big)^2 A_1 A_2,
\end{aligned}$$

$$\text{with} \quad A_1 = \left(1 + \|W_O\|\sqrt{H} \max_h \left[\|W^{V,h}\| \left[\frac{4}{\sqrt{d/H}}\|W^{Q,h}\|\|W^{K,h}\|+1\right]\right]\right)$$

$$A_2 = \left(1 + \|W_{FFN}\|\right)\|W_{\text{out}}\|$$

Let's now consider a perturbed input $\tilde{x} \in \mathcal{B}(x, \epsilon)$ as defined in Section 4.1. The previous upper-bound applies to any given point within that budget, and therefore we have:

$$\sup_{\tilde{x} \in \mathcal{B}(x,\epsilon)} d_{\mathcal{Y}}\left(\zeta_f\left(\tilde{x}\right), \zeta_f\left(x\right)\right) \le \mathcal{L}_f \epsilon$$

We can therefore conclude that, in respect of Definition 1, the head-only finetuning strategy is is $(\epsilon, \gamma_{\text{Head-Only}})$-*robust*, with:

$$\gamma_{\text{Head-Only}} = \left(\frac{d}{d-1}\right)^2 C_1 C_2 \epsilon,$$

$$\text{with: } C_1 = \left(1 + \|W_O\|\sqrt{H} \max_h \left[\|W^{V,h}\| \left[\frac{4}{\sqrt{d/H}}\|W^{Q,h}\|\|W^{K,h}\|+1\right]\right]\right),$$

$$C_2 = \left(1 + \|W_{FFN}\|\right)\|W_{\text{out}}\|,$$

$\square$

## B    PROOF OF THEOREM 1

**Theorem.** *Let $f: \mathcal{X} \to \mathcal{Y}$ be the pre-trained TBM-based model following the considered problem setup. For the LoRA-based finetuning strategy $\zeta_f^{LoRA}$, where the LoRA is only applied to the main transformer part, is $(\epsilon, \gamma_{LoRA})$-robust, with:*

$$\gamma_{LoRA} = \left(\frac{d}{d-1}\right)^2 C_1' C_2 \epsilon,$$

$$\text{with: } C_1' = 1 + \|W_O\|\sqrt{H} \max_h \left[\|W^{V,h}\| + \frac{\alpha}{r}\|A^{V,h}\|\|B^{V,h}\|\right]$$

$$\left[\frac{4}{\sqrt{d/H}}\left[\|W^{Q,h}\| + \frac{\alpha}{r}\|A^{Q,h}\|\|B^{Q,h}\|\right]\|W^{K,h}\|+1\right]$$

*Proof.* In this part, we considert the LoRA-based finetuning. In this perspective, we follow the same analogy as the previous proof. Specifically, Let $X \in \mathcal{X}$ be our input composed of $n$ tokens $x_i \in \mathbb{R}^d$. We consider the same model $f$ which is built using the dot-product self-attention as referred to in Equation 1 and reformulated as:

$$\text{AH}(x) = \text{Softmax}\left(\frac{(XW^Q)(XW^K)^T}{\sqrt{\frac{D}{H}}}\right)XW^V$$

$$= PXW^V = h(X)W^V,$$

We recall that in the case of LoRA, two additional matrices $A$ and $B$ are learnable during the finetuning. Specifically, given a weight matrix $W \in \mathbb{R}^{d \times k}$ in a model, it is substituted by the following:

$$W' = W + \frac{\alpha}{r}BA,$$

where $r$ is the rank of the low-rank adaptation, $\alpha$ is the scaling factor, and $B \in \mathbb{R}^{d \times r}$ and $A \in \mathbb{R}^{r \times k}$ are learnable weight matrices learned during the finetuning process.

From the previous section, we have the following Lipschitz bound for the head-only finetuning strategy:

$$\mathcal{L}'_f \leq L_{LN}^2 (1 + \mathcal{L}_{MH})(1 + L_{FFN})$$

$$\leq \left(\frac{d}{d-1}\right)^2 \left(1 + \mathcal{L}_{MH}\right)\left(1 + \|W_{FFN}\|\right)$$

$$\leq \left(\frac{d}{d-1}\right)^2 \left(1 + \|W_O\|\sqrt{H} \max_h \left[\|W'^{V,h}\|\left[\frac{4}{\sqrt{d/H}}\|W'^{Q,h}\|\|W'^{K,h}\|+1\right]\right]\right)\left(1 + \|W_{FFN}\|\right)$$

We consider that the LoRA finetuning is only applied to the main core of Transformer, specifically to the query $Q$ and value $V$ matrices of the attention mechanism as in the original work. We can therefore continue the previous computation by including the corresponding values:

$$\mathcal{L}'_f \leq \left(\frac{d}{d-1}\right)^2 \left(1 + \|W_O\|\sqrt{H} \max_h \left[\|W'^{V,h}\|\left[\frac{4}{\sqrt{d/H}}\|W'^{Q,h}\|\|W'^{K,h}\|+1\right]\right]\right)\left(1 + \|W_{FFN}\|\right)$$

$$\leq \left(\frac{d}{d-1}\right)^2 \left(1 + \|W_O\|\sqrt{H} \max_h \left[\left[\|W^{V,h}\|+\frac{\alpha}{r}\|A^{V,h}\|\|B^{V,h}\|\right]\right.\right.$$

$$\left.\left.\left[\frac{4}{\sqrt{d/H}}\left[\|W^{Q,h}\|+\frac{\alpha}{r}\|A^{Q,h}\|\|B^{Q,h}\|\right]\|W^{K,h}\|+1\right]\right]\right)\left(1 + \|W_{FFN}\|\right)$$

$$\leq \left(\frac{d}{d-1}\right)^2 A'_1 A_2,$$

Similar to the previous proof, let's consider a perturbed input $\tilde{x} \in \mathcal{B}(x, \epsilon)$ as defined in Section 4.1. The previous upper-bound applies to any given point within that budget, and therefore we have:

$$\sup_{\tilde{x} \in \mathcal{B}(x,\epsilon)} d_{\mathcal{Y}}\left(\zeta_f\left(\tilde{x}\right), \zeta_f\left(x\right)\right) \leq \mathcal{L}'_f \epsilon$$

We can therefore conclude that, in respect of Definition 1, the LoRA finetuning strategy is $(\epsilon, \gamma_{\text{LoRA}})$-*robust*, with:

$$\gamma_{\text{LoRA}} = \left(\frac{d}{d-1}\right)^2 C'_1 C_2 \epsilon,$$

$$\text{with: } C'_1 = 1 + \|W_O\|\sqrt{H} \max_h \left[\|W^{V,h}\|+\frac{\alpha}{r}\|A^{V,h}\|\|B^{V,h}\|\right]$$

$$\left[\frac{4}{\sqrt{d/H}}\left[\|W^{Q,h}\|+\frac{\alpha}{r}\|A^{Q,h}\|\|B^{Q,h}\|\right]\|W^{K,h}\|+1\right]$$

$\square$

## C    PROOF OF THEOREM 2

**Theorem.** *Let $f : \mathcal{X} \to \mathcal{Y}$ be our pre-trained TBM-based model. Let's consider the LoRA finetuning strategy, where all the low-rank matrices $A$ in layer $h$ are initialized as $A_0^{Q,h}$ (for Query) and $A_0^{V,h}$ (for Values), then the resulting $C'_1$ constant in $\gamma_{LoRA}$ ( Theorem 1) can be written as:*

$$C'_1 \leq K_1 \left(1 + \eta L\right)^t \max_h \|A_0^{(V,h)}\| + K_2 (1 + \eta L)^{2t} \max_h \|A_0^{(V,h)}\|\|A_0^{(Q,h)}\| + C,$$

*with $K_1, K_2$ and $C$ being the constants depending on the final weight norms (derived in Equation 12).*

*Proof.* Let's now consider the effect of Initialization distribution on the final adversarial robustness of the LoRA finetuning. Specifically, we consider the same settings as in prior work, where the $B$ matrix is set to 0 and only the $A$ matrix is initialized.

The gradient descent update at finetuning epoch $t$ for our matrix $A$ (at any layer) is written as:

$$A_{t+1}^{(Q,h)} = A_t^{(Q,h)} - \eta \nabla \mathcal{L}(A_t^{(Q,h)}).$$

As specified in our problem setup in Section 3, we consider that our loss function $\mathcal{L}$ to be $L$-smooth, we can hence write the following result:

$$\left\| \nabla \mathcal{L}(A_t^{(Q,h)}) \right\| \leq L \left\| A_t^{(Q,h)} - A_*^{(Q,h)} \right\|.$$

Consequently, after $t$ training epochs, we can write:

$$\begin{aligned}
\left\| A_t^{(Q,h)} \right\| &= \left\| A_{t-1}^{(Q,h)} - \eta \nabla \mathcal{L}(A_{t-1}^{(Q,h)}) \right\| \\
&\leq \left\| A_{t-1}^{(Q,h)} \right\| + \eta L \left\| A_{t-1}^{(Q,h)} - A_*^{(Q,h)} \right\| \\
&\leq (1 + \eta L) \left\| A_{t-1}^{(Q,h)} \right\| + \eta L \left\| A_*^{(Q,h)} \right\|.
\end{aligned}$$

In addition, we suppose that the considered learning rate is chosen as $\eta \leq \frac{1}{L}$. Consequently, we can write based on the previous formulation and by using recursion:

$$\left\| A_t^{(Q,h)} \right\| \leq (1 + \eta L)^t \left\| A_0^{(Q,h)} \right\| + \sum_{h=0}^{t} 2^h \left\| A_*^{(Q,h)} \right\| \tag{6}$$

$$\leq (1 + \eta L)^t \left\| A_0^{(Q,h)} \right\| + 2^{t+1} \left\| A_*^{(Q,h)} \right\| \tag{7}$$

$$\leq (1 + \eta L)^t \left\| A_0^{(Q,h)} \right\| + 2^{t+1} \left\| A^{(Q,h)} \right\| \tag{8}$$

*Remark.* We denote $A_*$ (which are the converged final weights) as $A$ directly to be inline with the previous theorems and results.

We additionally note that a similar analogy applies to the matrix $B$. Since we consider that specific matrix to be initialized to zero, the resulting terms of the initialization is therefore set to zero and only the final weight norm is seen in the upper-bound.

Consequently, and from the previous part, we had that the LoRA finetuning is is $(\epsilon, \gamma_{\text{LoRA}})$-*robust*, with:

$$\gamma_{\text{LoRA}} = \left( \frac{d}{d-1} \right)^2 C_1' C_2 \epsilon,$$

$$\text{with: } C_1' = 1 + \|W_O\| \sqrt{H} \max_h \left[ \|W^{V,h}\| + \frac{\alpha}{r} \|A^{V,h}\| \|B^{V,h}\| \right]$$

$$\left[ \frac{4}{\sqrt{d/H}} \left[ \|W^{Q,h}\| + \frac{\alpha}{r} \|A^{Q,h}\| \|B^{Q,h}\| \right] \|W^{K,h}\| + 1 \right]$$

Using the result from Equation 8, we can connect the previous resulting bound to the initialization, resulting in the following:

$$\text{with: } C_1' = 1 + \|W_O\| \sqrt{H} \max_h \left[ \|W^{V,h}\| + \frac{\alpha}{r} \|B^{V,h}\| \left[ (1 + \eta L)^t \left\| A_0^{(V,h)} \right\| + 2^{t+1} \left\| A^{(V,h)} \right\| \right] \right]$$

$$\left[ \frac{4}{\sqrt{d/H}} \left[ \|W^{Q,h}\| + \frac{\alpha}{r} \|B^{Q,h}\| \left[ (1 + \eta L)^t \left\| A_0^{(Q,h)} \right\| + 2^{t+1} \left\| A^{(Q,h)} \right\| \right] \right] \|W^{K,h}\| + 1 \right]$$

From the previous result, we can separate the two main terms, as follows:

$$C_1' = 1 + \|W_O\|\sqrt{H} \max_h \left\{ \underbrace{\left[ \|W^{V,h}\| + \tfrac{\alpha}{r}\|B^{V,h}\|\big((1+\eta L)^t\|A_0^{(V,h)}\| + 2^{t+1}\|A^{(V,h)}\|\big) \right]}_{=a_h} \right. \tag{9}$$

$$\times \underbrace{\left[ \tfrac{4}{\sqrt{d/H}}\big(\|W^{Q,h}\| + \tfrac{\alpha}{r}\|B^{Q,h}\|\big((1+\eta L)^t\|A_0^{(Q,h)}\| + 2^{t+1}\|A^{(Q,h)}\|\big)\big)\|W^{K,h}\| + 1 \right]}_{=b_h} \left. \vphantom{\right]} \right\}.$$

$$\tag{10}$$

The previous two elements can be written as:

$$a_h \leq C_h^{(V)} + \tfrac{\alpha}{r}\|B^{V,h}\|(1+\eta L)^t\|A_0^{(V,h)}\|$$

$$b_h \leq C_h^{(QK)} + \tfrac{4}{\sqrt{d/H}}\tfrac{\alpha}{r}\|B^{Q,h}\|(1+\eta L)^t\|A_0^{(Q,h)}\|\|W^{K,h}\|$$

Note that since everything is positive (given that the norms by definition are positive), then we have:

$$\max_h(a_h b_h) \leq \Big(\max_h a_h\Big)\Big(\max_h b_h\Big). \tag{11}$$

Consequently, we can write:

$$\max_h(a_h b_h) \leq K_1'(1+\eta L)^t\|A_0^{(V)}\|_{\max} + K_2'(1+\eta L)^{2t}\|A_0^{(V)}\|_{\max}\|A_0^{(Q)}\|_{\max} + C_0,$$

$$\text{with: } K_1' = \tfrac{\alpha}{r}\|B^V\|_{\max},$$

$$K_2' = \tfrac{4}{\sqrt{d/H}}\Big(\tfrac{\alpha}{r}\Big)^2\|B^V\|_{\max}\|B^Q\|_{\max}\|W^K\|_{\max},$$

and $C_0$ is a time-independent constant.

Let's define the following: $A_0^Q = \max_h\|A_0^{(Q,h)}\|$, and $A_0^V = \max_h\|A_0^{(V,h)}\|$.

Then we can finally write:

$$C_1' \leq K_1(1+\eta L)^t \max_h\|A_0^{(V,h)}\| + K_2(1+\eta L)^{2t}\max_h\|A_0^{(V,h)}\|\|A_0^{(Q,h)}\| + C,$$

with:

$$K_1 = \|W_O\|\sqrt{H}\tfrac{\alpha}{r}\|B^V\|_{\max}, \tag{12}$$

$$K_2 = 4\|W_O\|\tfrac{H}{\sqrt{d}}\Big(\tfrac{\alpha}{r}\Big)^2\|B^V\|_{\max}\|B^Q\|_{\max}\|W^K\|_{\max}. \tag{13}$$

$$\square$$

**Proof of Lemma 1:**

**Lemma.** *Consider LoRA matrices for each head $h = 1, \ldots, H$ initialized with entries drawn i.i.d. from $\mathcal{U}(-a, a)$ independently across heads and stacks. Then the expected value of the robustness constant $C_1'$ satisfies*

$$\mathbb{E}[C_1'] = \mathcal{O}\Big((1+\eta L)^{2t}a^2\big((\sqrt{r} + \sqrt{k}) + \sqrt{\log H}\big)^2\Big).$$

*Proof.* From the Theorem2, for sufficiently large $t$ we have

$$C_1' = \mathcal{O}\Big((1 + \eta L)^{2t} \max_h \|A_0^{(V,h)}\| \max_h \|A_0^{(Q,h)}\|\Big). \tag{14}$$

For a random matrix $A \in \mathbb{R}^{r \times k}$ with i.i.d. entries from $\mathcal{U}(-a, a)$, matrix concentration bounds yield

$$\mathbb{E}[\|A\|] \leq C_1 a(\sqrt{r} + \sqrt{k}), \tag{15}$$

for a universal constant $C_1 > 0$. Taking the maximum over $H$ independent heads gives

$$\mathbb{E}\left[\max_{h=1,\dots,H} \|A^{(h)}\|\right] \leq C_2 a\big((\sqrt{r} + \sqrt{k}) + \sqrt{\log H}\big). \tag{16}$$

By independence of Q and V stacks,

$$\mathbb{E}\left[\max_h \|A_0^{(V,h)}\| \max_h \|A_0^{(Q,h)}\|\right] = \mathbb{E}\left[\max_h \|A_0^{(V,h)}\|\right] \cdot \mathbb{E}\left[\max_h \|A_0^{(Q,h)}\|\right], \tag{17}$$

which yields the stated bound after plugging in Equation 14, multiplication and absorbing constants. $\square$

# D   ADDITIONAL RESULTS

## D.1   ADDITIONAL RESULTS CV

**Results SwiN-based Transformer.** While in Section 6, Figure 2 provides the results of the success rate of the ViT, we additionally provide the results on the SwiN, which show similar tendencies and insights on the link between loRA and adversarial robustness compared to head-only finetuning.

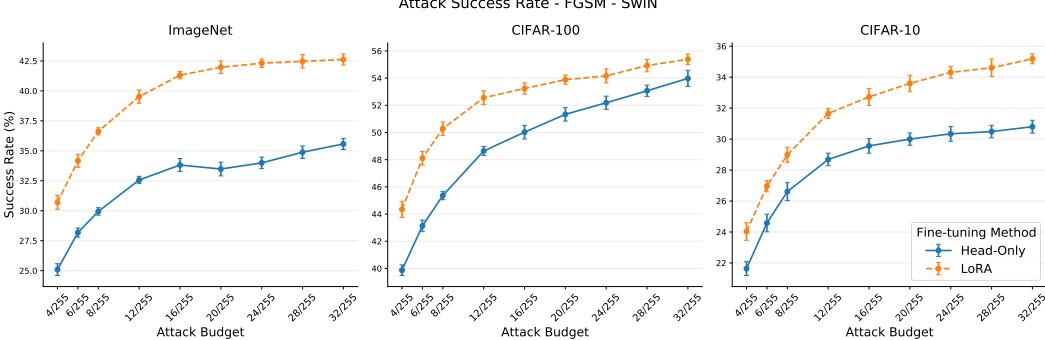

Figure 8: Success Rate of the FGSM Attack when applied to a SwiN-based Model and different datasets and attack budgets.

**Clean/Attacked Accuracy.** Additionally, as explained in Section 6, we study the adversarial robustness from an attack success rate perspective as a representative metric, but we additionally report the attacked accuracy, which also has the same insights. Note that the success rate is simply a representative function of the attack accuracy (without taking into account the erroneous predictions of the models). In this perspective, Figure 9 provides the results of the average clean and attacked accuracy of a ViT when subject to the FGSM attack for different attack budgets, while Figure 10 provides the results for the SwiN-based transformer.

**Other Results on Initialization.** In line with what was presented in Section 6.2, we extend the study to take into account the other considered datasets. In this perspective, Figure 11 provides the corresponding results on the ImageNet, the CIFAR-10 and CIFAR-100.

## D.2   ON THE EFFECT OF RANK

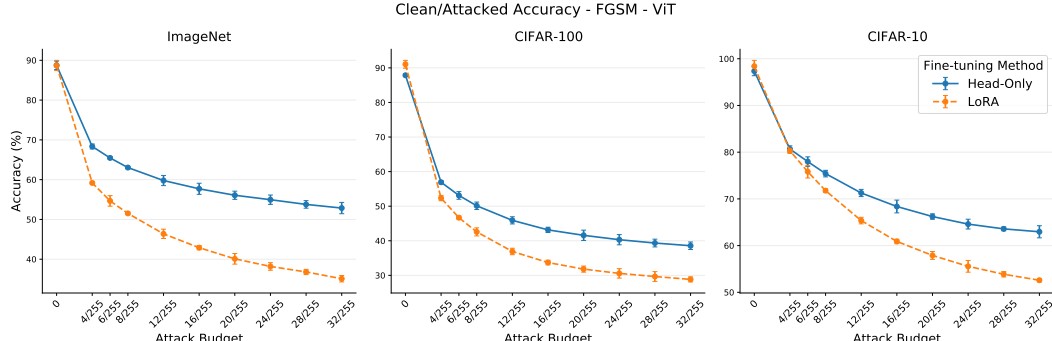

Figure 9: Clean/Attacked Accuracy of a ViT when subject to the FGSM attack for different budgets and datasets.

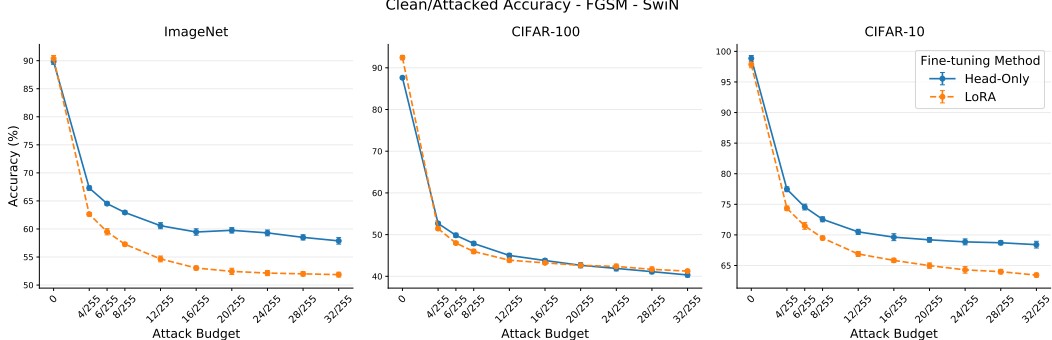

Figure 10: Clean/Attacked Accuracy of a SwiN-based model when subject to the FGSM attack for different budgets and datasets.

In addition to studying the effect of the $\alpha$ value in the empirical evaluation in Section 6 (Figure 6), we evaluate the effect of the chosen rank value $r$ which is also present in the computed upper-bound. Figure 12 shows the resulting study, where we see that increasing the value of the rank $r$ results in decreasing the attack success rate, in which is in accordance with our theoretical results, where $\gamma$ is inversely proportional to the chosen value $r$.

### D.3 ADDITIONAL RESULTS NLP

**Results for A2T Attack.** While in Section 6, Figure 3 presents the results of TextFooler attack on head-only and LoRA finetuning, Figure 13 presents the attack results from A2T attack. Overall, we have similar overall observations for the adversarial robustness comparison between head-only and LoRA finetuning.

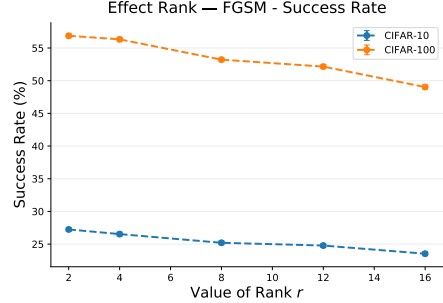

Figure 12: Effect of the rank value parameter $r$ of LoRA on the resulting attack success rate.

**Adversarial Robustness Comparison of Norm-Bounded LoRA.** While our theory show that LoRA is less robust than Head-Only finetuning, recent research work has also demonstrate that fine-tuning under bounded parameter norms improves the advesarial robustness of LoRA. Specifically, DeLoRA (Bini et al., 2025) normalizes and scales low-rank updates to decouple the direction of weight changes from their magnitude. More recently, NB-LoRA (Wang et al., 2025) reparameterizes low-rank adaptation matrices so that its singular vluaes are explicitly bounded, yielding improved training stability and hyper-parameter robustness compared to LoRA. Figure 14 shows the empirical comparision between LoRA, DeLoRA, NB-LoRA and Head-Only fine-tuning. In

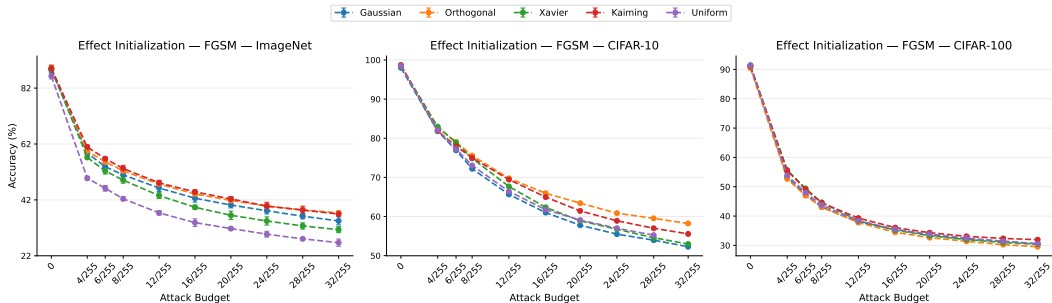

Figure 11: Clean/Attacked Accuracy of a ViT when subject to FGSM under different initialization distribution.

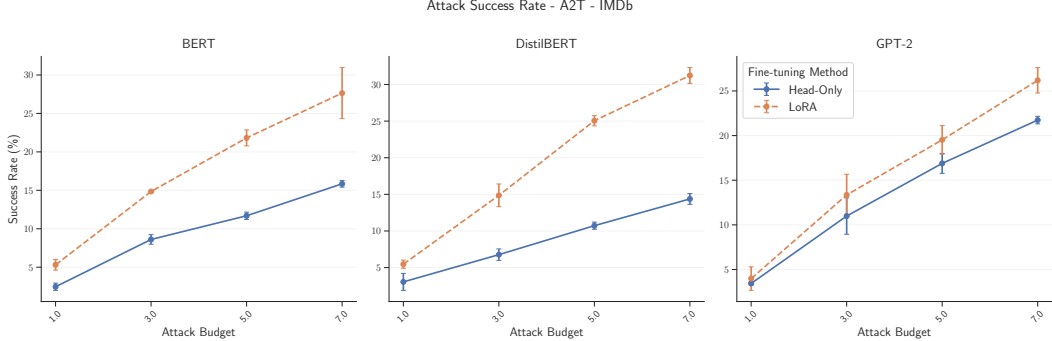

Figure 13: Attack Success Rate of the A2T Attack on BERT, DistilBERT, and GPT-2 when applied to IMDb dataset with different attack budget (number of words changed).

general, we find that both DeLoRA and NB-LoRA improves the adversarial robustness of the model as compared to LoRA.

**Clean/Attacked Accuracy.** In addition to attack success rate, Table 3 and 4 and provides detailed results of clean accuracy and adversarial accuracy of TextFooler and A2T attack.

Table 3: Original accuracy (Orig Acc), attack success rate (ASR, %) and adversarial accuracy (Adv Acc, %) across datasets with TextFooler attack. Attack budget: max number of words changed is 3.

| | Dataset | **IMDb** | | | **SST-2** | | | **Yelp Polarity** | | |
|---|---|---|---|---|---|---|---|---|---|---|
| | Metric | Orig Acc | ASR | Adv Acc | Orig Acc | ASR | Adv Acc | Orig Acc | ASR | Adv Acc |
| **BERT** | Head-Only | $83.20 \pm 0.87$ | $11.27 \pm 0.12$ | $71.93 \pm 0.83$ | $83.40 \pm 0.53$ | $54.00 \pm 0.00$ | $29.40 \pm 0.53$ | $86.13 \pm 1.53$ | $14.73 \pm 2.16$ | $71.40 \pm 0.72$ |
| | LoRA | $90.53 \pm 0.92$ | $16.67 \pm 1.01$ | $73.87 \pm 1.89$ | $92.13 \pm 0.23$ | $53.40 \pm 1.64$ | $38.73 \pm 1.86$ | $92.60 \pm 0.53$ | $15.93 \pm 0.81$ | $76.67 \pm 1.22$ |
| **DistilBERT** | Head-Only | $83.60 \pm 0.40$ | $11.40 \pm 0.35$ | $72.20 \pm 0.72$ | $82.13 \pm 0.31$ | $52.00 \pm 0.72$ | $30.13 \pm 0.50$ | $86.07 \pm 1.86$ | $14.60 \pm 1.51$ | $71.47 \pm 0.70$ |
| | LoRA | $90.07 \pm 0.31$ | $17.53 \pm 1.22$ | $72.53 \pm 1.10$ | $89.47 \pm 0.83$ | $58.00 \pm 0.69$ | $31.47 \pm 0.92$ | $92.07 \pm 0.42$ | $17.93 \pm 3.60$ | $74.13 \pm 3.92$ |
| **GPT-2** | Head-Only | $85.67 \pm 1.01$ | $6.73 \pm 0.81$ | $78.93 \pm 1.72$ | $82.07 \pm 0.50$ | $47.20 \pm 0.92$ | $34.87 \pm 0.76$ | $85.20 \pm 1.60$ | $10.07 \pm 0.81$ | $75.13 \pm 2.34$ |
| | LoRA | $91.87 \pm 0.90$ | $7.87 \pm 0.42$ | $84.00 \pm 0.69$ | $90.93 \pm 1.33$ | $54.60 \pm 2.11$ | $36.33 \pm 2.02$ | $92.40 \pm 0.53$ | $7.87 \pm 0.81$ | $84.53 \pm 0.99$ |
| **Gemma-2B** | Head-Only | $92.13 \pm 0.58$ | $7.73 \pm 0.64$ | $84.40 \pm 1.22$ | $88.73 \pm 0.46$ | $46.80 \pm 0.20$ | $41.93 \pm 0.50$ | $94.87 \pm 0.23$ | $7.80 \pm 1.06$ | $87.07 \pm 1.15$ |
| | LoRA | $94.40 \pm 0.53$ | $10.80 \pm 0.53$ | $83.60 \pm 0.69$ | $96.07 \pm 0.31$ | $40.60 \pm 1.73$ | $55.47 \pm 1.70$ | $97.13 \pm 0.50$ | $9.40 \pm 1.51$ | $87.73 \pm 1.81$ |
| **LLaMA-3.2-1B** | Head-Only | $93.33 \pm 0.81$ | $7.13 \pm 0.81$ | $86.20 \pm 1.59$ | $86.73 \pm 0.12$ | $47.60 \pm 0.53$ | $39.13 \pm 0.64$ | $94.53 \pm 0.31$ | $10.27 \pm 0.81$ | $84.27 \pm 0.99$ |
| | LoRA | $93.33 \pm 0.23$ | $8.67 \pm 0.42$ | $84.47 \pm 0.42$ | $95.93 \pm 0.61$ | $43.93 \pm 1.79$ | $52.00 \pm 1.22$ | $96.80 \pm 0.92$ | $8.73 \pm 1.10$ | $88.67 \pm 0.64$ |
| **Mistral-7B** | Head-Only | $93.40 \pm 0.53$ | $6.47 \pm 0.50$ | $86.93 \pm 0.95$ | $91.00 \pm 0.53$ | $43.93 \pm 0.58$ | $47.07 \pm 1.01$ | $96.53 \pm 1.55$ | $7.60 \pm 0.72$ | $88.93 \pm 0.99$ |
| | LoRA | $89.40 \pm 6.05$ | $12.40 \pm 7.50$ | $77.00 \pm 13.53$ | $81.80 \pm 2.12$ | $58.33 \pm 3.75$ | $23.47 \pm 1.75$ | $97.33 \pm 0.61$ | $7.07 \pm 1.68$ | $90.27 \pm 1.14$ |
| **TinyLLaMA** | Head-Only | $93.60 \pm 0.92$ | $5.53 \pm 0.31$ | $88.07 \pm 0.81$ | $73.33 \pm 0.12$ | $45.60 \pm 0.20$ | $27.73 \pm 0.31$ | $89.73 \pm 1.86$ | $13.33 \pm 3.01$ | $76.40 \pm 3.82$ |
| | LoRA | $93.13 \pm 1.03$ | $10.87 \pm 0.42$ | $82.27 \pm 1.40$ | $93.80 \pm 0.20$ | $49.00 \pm 1.59$ | $44.80 \pm 1.64$ | $95.33 \pm 1.22$ | $13.13 \pm 1.63$ | $82.20 \pm 2.78$ |

# E EXPERIMENTAL DETAILS

**Computer-Vision.** For all the experiments, we used a learning rate of $1e-03$ to train the LoRA and the head-only finetuning, and the training was done using AdaM optimizer (Kingma & Ba, 2014).

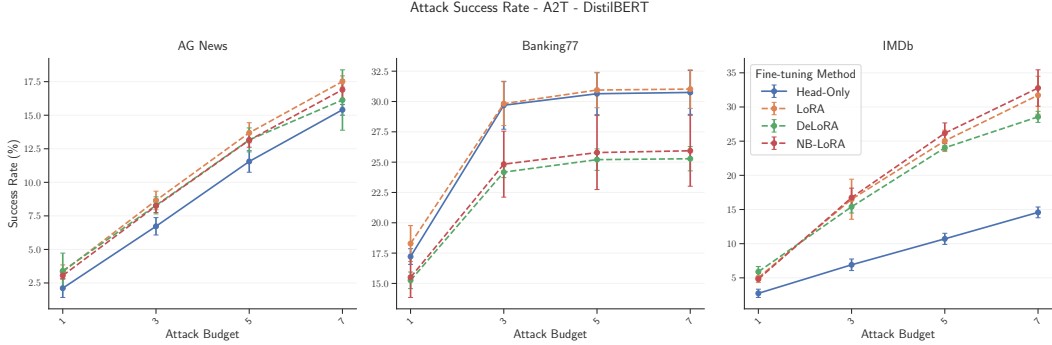

Figure 14: Attack Success Rate of the A2T Attack on DistilBERT model when applied to AG News, Banking77, IMDb dataset with different attack budget (number of words changed).

Table 4: Original accuracy (Orig Acc), attack success rate (ASR, %) and adversarial accuracy (Adv Acc, %) across datasets with A2T attack. Attack budget: max number of words changed is 3.

| | Dataset | **IMDb** | | | **SST-2** | | | **Yelp Polarity** | | |
|---|---|---|---|---|---|---|---|---|---|---|
| | Metric | Orig Acc | ASR | Adv Acc | Orig Acc | ASR | Adv Acc | Orig Acc | ASR | Adv Acc |
| **BERT** | Head-Only | $86.67 \pm 0.92$ | $8.61 \pm 0.62$ | $79.20 \pm 0.40$ | $83.40 \pm 0.53$ | $28.13 \pm 0.74$ | $59.93 \pm 0.23$ | $87.00 \pm 1.06$ | $11.04 \pm 1.13$ | $77.40 \pm 1.93$ |
| | LoRA | $92.93 \pm 1.01$ | $14.85 \pm 0.06$ | $79.13 \pm 0.81$ | $92.13 \pm 0.23$ | $22.86 \pm 0.62$ | $71.07 \pm 0.46$ | $93.20 \pm 0.72$ | $11.52 \pm 0.70$ | $82.47 \pm 1.14$ |
| **DistilBERT** | Head-Only | $87.67 \pm 0.31$ | $6.77 \pm 0.78$ | $81.73 \pm 0.58$ | $82.13 \pm 0.31$ | $28.33 \pm 0.40$ | $58.73 \pm 0.23$ | $86.40 \pm 1.74$ | $11.26 \pm 0.74$ | $76.60 \pm 1.04$ |
| | LoRA | $92.87 \pm 0.31$ | $14.86 \pm 1.55$ | $79.07 \pm 1.70$ | $89.47 \pm 0.83$ | $27.80 \pm 1.71$ | $64.60 \pm 1.91$ | $93.33 \pm 1.03$ | $17.50 \pm 2.00$ | $76.93 \pm 2.05$ |
| **GPT-2** | Head-Only | $89.27 \pm 0.95$ | $10.99 \pm 2.04$ | $79.47 \pm 2.39$ | $82.07 \pm 0.50$ | $30.06 \pm 0.42$ | $57.40 \pm 0.40$ | $85.80 \pm 1.91$ | $12.29 \pm 2.30$ | $75.27 \pm 3.06$ |
| | LoRA | $93.93 \pm 0.64$ | $13.35 \pm 2.32$ | $81.27 \pm 2.52$ | $90.93 \pm 1.33$ | $22.00 \pm 0.77$ | $70.93 \pm 1.62$ | $92.73 \pm 0.50$ | $8.85 \pm 1.37$ | $84.53 \pm 1.62$ |

All the tasks were trained for 5 epochs; all of which yielded stable convergence. All the experiments were run 3 times with different seeds to reduce the effect of randomness, and we report the average and the corresponding standard deviations. For the LoRA model, we use the adaptation for all the weights within the self-attention mechanism.

After the convergence of each finetuning strategy, and as explained in Section 6, we used the FGSM and PGD attack with different budget attacks. For the PGD, we set the number of iterations to $10$, which is in line with the literature. For the attack budget, we used different attack budgets to illustrate the effect of the attack. Specifically we use a range varying from $4/255$ to $32/255$. We note that the classical budget for CIFAR-10 and CIFAR-100 is considered to be $8/255$, while for the ImageNet it's set to be $4/255$.

Finally, for the DeLoRA, we set the $\lambda$ parameter to $15$, following the original work which suggested a value between $10$ and $20$.

**Natural-Language-Processing.**

For both head-only and LoRA fine-tuning, we take the pretrained checkpoint of the model and further finetune it for 5 epochs using the AdamW (Loshchilov & Hutter, 2019) optimizer with a learning rate of $2e-4$ and batch size of $16$. We use the HuggingFace Datasets[1] to get the dataset for the experiment. For the Yelp Polarity dataset, we subsample to 3000 train instances, 1000 validation instances, and 1000 test instances to speed up the training process. For IMDb and SST-2 datasets, we use the full dataset and the original train, validation, and test splits provided by Huggingface.

After the fine-tuning phase, we subsample $500$ instances from the test split and run both the TextFooler and A2T attack. The subsampling process is seeded to have a more statistically sound evaluation of the adversarial robustness of head-only and LoRA finetuning. For the TextFooler attack, we allow at most 10 word substitutions per example. Candidate substitutions are drawn from counter-fitted word embeddings (with pre-computed cosine similarity matrix), with up to 50 candidates per word, and are filtered using a sentence-level similarity threshold of $0.7$. During model inference for the attack, we use a batch size of 16. For the A2T attack, we allow for the number

---

[1]https://huggingface.co/docs/datasets/en/index

of word changes across the set $\{1, 3, 5, 7\}$ and a query budget of at most $50$ forward passes per example. Up to $20$ candidate substitution words are allowed for each word in a sentence, and the substitution words are further narrowed with a word-level similarity of $0.8$. Finally, a sentence level similarity of $0.9$ is used to filter the final sentence. The model inference during the attack is run with a batch size of $32$.

All experiments are repeated 3 times with different seeds, reporting the average performance together with standard deviations.

