# OpenReview forum: "Theoretically Understanding the Hidden Adversarial Price of Low-Rank Adaptation"
_ICLR.cc/2026/Conference — Submitted to ICLR 2026_

### Official Review · Reviewer_zHsT · 2025-10-25

**Soundness:** 2
**Presentation:** 2
**Contribution:** 1
**Rating:** 4
**Confidence:** 4

**Summary:**

This work investigates the adversarial robustness of head-only and LoRA fine-tuning methods. A theoretical upper bound on robustness is derived, and empirical attack results on ViT and LLM fine-tuning demonstrate that the head-only method exhibits stronger robustness than LoRA.

**Strengths:**

1. Adversarial robustness is an interesting topic as fine-tuning may lead to over-fitting due to small datasets in downstream tasks.

**Weaknesses:**

1. It is not surprising that head-only tuning is more robust than LoRA. When viewed within the full fine-tuning framework, the head-only approach can be seen as applying sparse gradients (i.e., dropping all weight gradients except those of the head), whereas LoRA applies low-rank but still dense gradient updates. Consequently, head-only tuning imposes a much stronger form of regularization than LoRA.

2. The theoretical bound estimates in Proposition 1 and Theorem 1 appear to be quite conservative. Specifically, for Thm.1 it is possible to have $\\|AB\\|=0$ (i.e., LoRA has no effect on the base model) while $\\|A\\|\cdot \\|B\\|$ is arbitrarily large. For example, consider $A=\begin{bmatrix} a & 0 \end{bmatrix}$ and $B=\begin{bmatrix} 0 \\\\ b^\top \end{bmatrix}$ with $a,b\in \mathbb{R}^n$ having large norms. Similar examples can be constructed for $W^Q W^K$ in Prop.1 as well. There exist much tighter Lipschitz bound estimation methods in the literature, such as [1] for the attention layer and [2] for the MLP layer.

3. The robustness bound estimation also seems disconnected from the empirical findings reported in this work. As I understand it, the theoretical analysis suggests that fine-tuning under bounded parameter norms may enhance robustness. The argument would be more convincing if supported by corresponding empirical evidence. Recent studies [3, 4] have proposed LoRA variants with explicit norm-bounding guarantees that could serve as relevant comparisons.

[1] Havens et al. Fine-grained Local Sensitivity Analysis of Standard Dot-Product Self-Attention. ICML 2024

[2] Wang et al. On the Scalability and Memory Efficiency of Semi-definite Programs for Lipschitz Constant Estimation of Neural Networks: Scaling the Computation for ImageNet. ICLR 2024.

[3] Bini et al. DeLoRA: Decoupling angles and strength in low-rank adaptation. ICLR 2025.

[4] Wang et al. Norm-bounded Low-Rank Adaptation. arXiv 2501.19050.

**Questions:**

1. For the bound estimation, why take the maximum operator over the heads (i.e. $\mathrm{max}_h$)?

2. The FFN layer is not introduced, i.e., where does $W_{FFN}$ come from?

---

> ### Author Response · Authors · 2025-11-26
>
> We appreciate the reviewer's detailed feedback, which has been instrumental in improving our work. Several raised points highlighted areas where additional clarity was needed, and we have worked to address these in the revised manuscript. Below, we provide responses to the specific concerns along with corresponding revisions.
>
> **Q1-Regarding the maximum term in the bound:** We thank the reviewer for pointing this out to us and for their interest in the theoretical proof. The maximum aspect comes from the fact that after computing the bound for each self-attention head, we need to concatenate them to represent the final norm of the overall attention block. From this perspective, Max represented a good approach to achieving a tighter bound.
>
> **Q2 - Regarding the FFN Layer:** We apologize if this wasn’t emphasized enough in our manuscript, but in the preliminaries section (Line 140 - 141), we provided elements regarding the considered section. Specifically, the FFN in this case is part of the classical vision architecture, which we consider in our theoretical analysis.
>
> **W1/W2-On the work’s main idea:**
> We thank the reviewer for this thoughtful comment, and we agree with the reviewer’s assessment that, at a high level, one can indeed view head-only finetuning as a more strongly regularized procedure than LoRA: only the final prediction head is updated, whereas LoRA introduces low-rank but dense updates inside the backbone. Our empirical evaluation results are indeed consistent with this intuition.
>
> Our work’s main aim in this perspective is to formalize theoretically such an effect and provide theoretical elements on its existence. In addition, our theoretical analysis resulted in a better understanding of the existence of this gap and showed its high sensitivity to some specific design choices, e.g., $\alpha$, rank, initialization, in a way that is not fully captured by a “sparse vs. dense gradients” story.
>
> Specifically, our contributions go beyond the qualitative intuition in the following ways:
>
> - *Formal robustness bounds for head-only and LoRA finetuning:* We derive explicit adversarial risk bounds for head-only and LoRA, related to the model’s architecture and norm weights (and opening the door to other variants to benefit from this bound)
> - *Dependence on LoRA design choices:* Our theory links robustness to $\alpha$, rank, and initialization scale, giving therefore insights into small choices that could greatly enhance robustness in a “free-lunch” aspect.
> - *Systematic cross-architecture and cross-modal evidence:* We provide a broad empirical study across vision (ViT/Swin on CIFAR-10/100 and ImageNet-100) and language (BERT, DistilBERT, GPT-2, Gemma, LLaMA, Mistral) under multiple attacks, validating our theoretical contribution.
>
>
> **W3 - On Norm-Bounding LoRA variants:** We thank the reviewer for their insightful comment. While in the original manuscript, we focused on empirically validating the insights of the gap head-only/LoRA robustness, we agree with the reviewer that, actually, our theoretical analysis is the proof that norm-bounded LoRA variants could be very useful in the literature and may have an impact on enhancing the model’s robustness. In this perspective, and in line with the reviewer’s suggestion, we have added some experimental results on this aspect, where we consider DeLoRA (Figure 7) and NB-LoRA (Figure 14 - Appendix D.3), where we show that enforcing parameter-norm constraints indeed improves adversarial robustness, which is consistent with our theoretical prediction. We have added a specific section in the updated Manuscript (Section 6.3), in which we further consider discussing this perspective.

---

> > ### Comment · Reviewer_zHsT · 2025-11-27
> >
> > Thank you for further explanation and efforts on the new robustness results, which fits very well with the theoretical intuition.
> >
> > I will keep my score since 1) similar theoretical results can be found in the literature of Lipschitz neural networks and transformers; 2) Even with large pretrain dataset and computational resource, the adversarial robustness might still be an issue for those foundation models. When fine-tuning with small amount downstream task data, it is not surprising that robustness become worse.

---

> > > ### Author Response · Authors · 2025-11-28
> > >
> > > Thank you very much for the thoughtful follow-up and for taking into account our rebuttal and our effort to clarify some elements. We sincerely appreciate the effort and consideration.
> > >
> > > Regarding the first point, while we agree that prior work has analyzed Lipschitz properties of self-attention and Transformer architectures, our contribution is distinct in that we study how different parameter-efficient fine-tuning strategies affect adversarial robustness, specifically comparing LoRA to head-only tuning. To the best of our knowledge, this comparison—and the resulting theoretical gap—has not been established in prior literature. This gap also helps explain why recently proposed variants such as NB-LoRA or DeLoRA may display improved robustness. Furthermore, we take an additional step by examining how LoRA-specific design choices (initialization, rank, scaling $\alpha$) influence the derived bounds, offering actionable insights for practitioners. We therefore believe our work provides a new and complementary perspective to existing analyses of Lipschitz continuity in Transformers.
> > >
> > > On the second point, we would like to clarify that our study is orthogonal to the robustness of the pre-trained backbone. Our setting assumes a pre-trained model is already fixed, and our goal is to understand the inference-time robustness implications of different fine-tuning mechanisms, especially in realistic scenarios where downstream data is limited. As identified by the reviewer, fine-tuning on small datasets can indeed exacerbate instability, which further motivates the need to theoretically understand how different adaptation choices affect robustness. While connecting these two directions of pre-training and the expected robustness during finetuning seems an interesting direction, we consider that it’s out of our scope.
> > >
> > > We would be very happy to further clarify any aspect of the work, and we would be sincerely grateful if the reviewer could kindly reconsider the novelty and value of the contributions in light of these clarifications.

---

### Official Review · Reviewer_jokf · 2025-10-30

**Soundness:** 3
**Presentation:** 2
**Contribution:** 2
**Rating:** 4
**Confidence:** 3

**Summary:**

This paper investigates the relationship between LoRA fine-tuning strategies and adversarial robustness. The authors formalize the notion of expected adversarial robustness and theoretically show that, compared to head-only fine-tuning, LoRA’s gains in clean accuracy come at the cost of increased vulnerability to adversarial attacks. They further analyze how the initialization scheme impacts LoRA’s adversarial robustness.

**Strengths:**

1. This paper is well written and clearly structured.
2. The derivations of the proposition and theorems are detailed and well-explained, which facilitates understanding.
3. The theoretical findings are supported by extensive experiments on both vision and NLP tasks.

**Weaknesses:**

1. Several claims lack proper citations. For example, in Section 1 (Introduction), the sentence starting with “A common approach is to freeze ...” (line 17), and the discussion of evasion attacks in Section 4.1 (Adversarial Robustness) require citations.
2. In the supplementary material (Section B, Proof of Theorem 1), the last sentence:“..., head-only finetuning strategy”, appears to be a typo. It should be the LoRA fine-tuning strategy.
3. The term “consistently” is used when explaining experimental results. However, Table 2 shows that on SST-2 and Yelp Polarity datasets, LoRA outperforms Head-Only for both BERT and GPT-2.
4. In Theorem 2 and Lemma 1, the symbol h is used for both layer and head, which may lead to confusion.
5. The hyperparameters used in different adversarial attacks should be listed in the supplementary material for reproducibility.

**Questions:**

1. If LoRA is further applied to the K and O projection matrices, would it affect the current conclusions? How would C_1^\prime change in this case? A brief derivation or at least an experiment would be helpful.
2. According to Theorem 1, the rank also influences C_1^\prime. Could the authors provide experimental results exploring the impact of different rank values?

---

> ### Author Response · Authors · 2025-11-26
>
> We thank the reviewer for the thorough review and the raised questions. We are grateful to the reviewer for acknowledging the worth of our theoretical analysis and our empirical validation. We are also very thankful for all the elements and questions that help increase the quality of our manuscript. In what follows, we aim to address some of the weaknesses raised by the reviewer and clarify some elements to shed light on the overall perspective of our work.
>
> **Q1 - On the extension of LoRA to $K$ and $O$ :**  This is indeed a very relevant question. Basically, in our theoretical perspective, we only focused on applying LoRA to one part of the attention head; nonetheless, the results can be extended easily to the general use case where all the weights are subject to LoRA finetuning. In this perspective, the final $\gamma’$ will include an additional term for each individual weight, in which we can see the additional corresponding $A$ and $B$ matrices. We note that in our experimental setting, we apply the LoRA finetuning to all the layers within the self-attention framework. We have added some elements to clarify this in the experimental setting, and we thank the reviewer for pointing this out.
>
> **Q2 - On the effect of LoRA rank on C_1^\prime in Theorem 1:** We thank the reviewer for this suggestion. In the revised version, we have added an ablation on the LoRA rank, which is reported in Figure 12 in the appendix. In this experiment, we fix all training and attack settings and vary the rank $r$ of LoRA. As summarized in the paper, increasing the rank $r$ consistently decreases the attack success rate (yielding higher adversarial robustness of the model). This empirical observation matches our theoretical result in Theorem 1, where the robustness bound of LoRA is inversely related to the chosen rank $r$ via its contribution to C_1^\prime.
>
> **W1 - Missing citations in Intro and Sec. 4.1:** We appreciate the reviewer pointing this out and have revised the manuscript accordingly. In Section 1 (Introduction), for the sentence starting with “A common approach is to freeze ...”, we now cite representative works on freezing large models and training a downstream classification head only. In Section 4.1 (Adversarial Robustness), we have added canonical references for evasion attacks.
>
> **W2 - Last sentence in Appendix B:** We thank the reviewer for catching this typo. In the revised supplementary material, we have corrected “head-only finetuning strategy” to “LoRA finetuning strategy” to accurately reflect the setting analyzed in the theorem.
>
>
> **W3 - Use of “consistently” in describing Table 2 results:** We thank the reviewer for the comment and agree that “consistently” is too strong. In the revised manuscript, we now rephrase to “LoRA fine-tuning generally shows lower robustness compared to head-only tuning.” This updated phrasing better reflects the empirical results and aligns them with our theoretical discussion.
>
> **W4 - The use of symbol $h$ in Theorem 2 and Lemma 1:”**
> We thank the reviewer for pointing out this ambiguity. In the revised version, we have disentangled the notation by using distinct symbols for layer and head indices (e.g., using h for layers and n for heads). We have updated the statements of Lemma 1, as well as their proofs and surrounding text, to ensure that each symbol has a unique meaning and the notation is now unambiguous.
>
> **W5 - Reporting hyperparameters of adversarial attacks:** We apologize if the experimental details weren't exhaustive enough in the original manuscript. We totally agree with the reviewer on the importance of these elements, and we have added details of hyperparameters in the supplementary material for the adversarial attacks used in our experiments in the updated manuscript.

---

> ### Comment · Reviewer_jokf · 2025-11-27
>
> Thank you for the detailed explanation. I have two brief follow-up questions:
>
> 1. Regarding the Q1 answer, the phrase "applying LoRA to one part of the attention head" has made me more confused. Do you mean that LoRA is applied to a subset of attention heads, or is it applied only to specific projection matrices (e.g., Q and V, or some subset of {Q K V O}) within each head?
>
> 2. Since we do not have access to the revised version, would it be possible for you to briefly summarize or show the main text you plan to add (e.g., the clarification in the experimental setting, the ablation on LoRA rank, and the attack hyperparameters)? A short outline of those additions would help me better see how the revision addresses the concerns.
>
> Thank you again for the careful response. Really appreciated.

---

> > ### Author Response · Authors · 2025-11-28
> >
> > We thank the reviewer for engaging in the discussion and the follow-up questions. We are really grateful for the dedicated time to both reviewing the paper and interacting with our rebuttal.
> >
> > **Regarding Q1:** We apologize if this wasn’t clear enough, but there are two direction to consider here:
> >
> > - *In the theoretical results:* We consider that LoRA is only applied to the $V$ and $Q$ part of the attention (whose weights are denoted as $W^{V,h}$ and $W^{Q, h}$ in the theoretical derived bounds. Shall choice could be easily also extended to the other component within each head such as $K$ and $O$. The change in the $\gamma’$ will just reflect some additional weight norms of $A$ and $B$ related to these additional components.
> >
> > - *From an empirical perspective:* During the experiments, we consider that the LoRA is applied to all the components within each attention head of our model.
> >
> > We apologize for the confusion that may have resulted from our original answer, and we are happy to further clarify if needed.
> >
> > **Q2 - Regarding the additional results and experiments:** We wanted first to clarify that actually the reviewer should have access to the revised manuscript as we have uploaded a new version during the rebuttal (which is allowed by ICLR and we checked through a private window where we see the changes reflected). Could you please check again the pdf in OpenReview ?
> >
> > Nonetheless, we can summarize the edits that we did as follows:
> >
> > - We considered a range of different values of the rank $r$ for both CIFAR-10 and CIFAR-100 and then consider the attack success rate. We experimentally see that when increasing the value of the rank $r$, the corresponding success rate decreases, in which is in accordance with our theoretical results, where $\gamma$ is inversely proportional to the chosen value $r$. Figure 12 in Appendix D.2 shows the results of the study.
> >
> > - Regarding the attack parameters: We added specific elements regarding the Vision-related attacks (basically PGD number of iterations and the considered range of $\epsilon$ budget). For the NLP-related tasks, we provide elements regarding the parameters used for TextFooler and A2T attacks (such as the number of words allowed to change - reflecting the attack budget and the different consider query budgets).
> >
> > - Regarding formatting elements: We have tried to adapt based on the reviewer’s original review, where we agree that some notations were confusing and therefore disentangled some double occurrences and using distinct symbols for better clarity.
> >
> > We are really grateful for the reviewer for all the insightful comments and directions, which clearly enhanced the quality of our manuscript. We are happy to further clarify any element if need.

---

### Official Review · Reviewer_8iVG · 2025-11-04

**Soundness:** 2
**Presentation:** 3
**Contribution:** 2
**Rating:** 4
**Confidence:** 4

**Summary:**

This paper investigates the adversarial robustness of low-rank adaptation (LoRA). The authors use theoretical and empirical results to validate that the model after LoRA is more adversarially vulnerable compared to the model after the head-only tuning. Besides, they show that the parameter initialization with a smaller norm and the smaller scale factor can enhance robustness.

**Strengths:**

- Provide some good insights about the adversarial robustness after LoRA from theoretical results. The robustness is affected by parameter initialization and the scaling factor.
- Provide comprehensive results using CV and NLP datasets to support the claim.

**Weaknesses:**

- I am sceptical about the fairness of the comparison between LoRA and head-only fine-tuning. These two types of fine-tuning methods utilize different amounts of trainable parameters for fine-tuning. LoRA naturally introduce more parameters compared to head-only fine-tuning. Thus, LoRA naturally has a larger parameter space that can be threatened by the adversary. Therefore, I think the comparison is somewhat meaningless.
- Regarding the experiments of parameter initialization: The authors should show the different norms of parameters w.r.t. the adversarial robustness. Besides, they should show the compatibility of parameter initialization with AutoLoRA and ADV-LoRA.
- Seems that making the scaling factor smaller is equivalent to making the effect of the LoRA module smaller on the performance according to the Eq of LoRA, which is a trivial idea.

**Questions:**

Please see the Weaknesses.

---

> ### Author Response · Authors · 2025-11-26
>
> We thank the reviewer for the thorough review and the raised questions. We are specifically grateful for acknowledging the contribution from a theoretical and empirical perspective. In what follows, we aim to address some of the weaknesses raised by the reviewer and clarify some elements to shed light on the overall perspective of our work.
>
> **W1 - Regarding the fairness of the comparison:**
> We thank the reviewer for raising this point, which is indeed a relevant and fair point to discuss. In practice, when dealing with large models, the final user is usually approaching the adaptation of the pre-trained models through a perspective of fine-tuning with different strategies being available, such as head-only tuning and Low-Rank adaptation. These methods, as identified by the reviewer, don't have the same number of parameters (and the same degrees of liberty); nonetheless, it's because of this difference that a specific user would choose one or another (depending on the desire to perform the best clean accuracy). Our work, therefore, is around this choice, where we claim (and theoretically and empirically validate) that such a choice may come with a price of adversarial robustness.
> Beyond this remark, our work establishes an upper-bound that connects such effects to different choices, such as Initialization and the parameters of the LoRA. Therefore, we provide elements to increase the gap between the LoRA (which gives the best clean accuracy) and the head-only (which would give the best robustness).
> Furthermore, our work provides insights on where other variants of LoRA could intervene (such as reducing the norm of $A$ and $B$ and their connection to decreasing $\gamma$ - for instance, adaptation such as DeLoRa - Figure 7).
>
>
>
> **W2 - On the Norm of the Initialization:** We thank the reviewer for pointing out this direction. In Theorem 2, the theoretical link between increasing the norm of the initial weights and having a bigger $\gamma$ is clear. In line with the reviewer’s proposition, we have included a study showcasing empirically such an effect in Figure 6, where we consider different scaling values of the initial weight and see that it results in different underlying robustness.
>
> Regarding the generalization of our insights to the case of AutoLora and AdvLoRa, we would like to note that in the theoretical aspect, we don’t assume any element regarding the weights at epoch $t$ (either coming from the vanilla training of a LoRA or other adaptation of adversarial training), and therefore we would assume that the results can easily be extended to this line of adaptation. We are working on extending the results to the AutoLoRA, which is taking some time since the original code was adapted to the ResNet, and our adaptation doesn’t really match the results reported by the authors in their rebuttal regarding the ViT, necessitating therefore a better parameter tuning search.
>
> **W3 - On the intuitiveness of the results:** Indeed, the idea of scaling the effect of LoRA (through $\alpha$) may seem intuitive, but the main idea of the theoretical study is to directly connect the intuition to a theoretical framework that is measurable, showcasing the effect of this choice in the final robustness. Specifically, the main insight derived from this part of the theoretical and empirical evaluation is that changing the value of $\alpha$ (and the rank $r$ actually as provided in Figure 12, Appendix D.2) can have a great impact on increasing the robustness, and therefore should be carefully chosen.

---

### Official Review · Reviewer_AbCa · 2025-11-06

**Soundness:** 3
**Presentation:** 3
**Contribution:** 3
**Rating:** 6
**Confidence:** 1

**Summary:**

This paper investigate an under-explored region: the relationship between fine-tuning strategies and adversarial robustness.
More concretely, it formalizes the notion of expected adversarial robustness uppper bounds in a theoretical way, and analyze the trade-off of PEFT efficiency and adversarial robustness between two main-stream model tuning method, e.g. LoRA and SFT.
The paper further explores different LoRA initialization schemes aimed at improving robustness.
The demonstrating experiments are extensive, ranging from multiple vision and language benchmarks, but there still lack fundamental settings to fully justify the conclusion, e.g. more loose assumption but not the bounded activation, the introduction of adaptive defense, and the relationship between adversarial robustness and other PEFT methods, such as Prompt Tuning, Visual Prompt Tuning, etc.
Empirical evaluations are extensive across both vision and language benchmarks; however, several fundamental settings remain missing to fully justify the conclusions—for example, relaxing the overly strong bounded-activation assumption, introducing adaptive defense baselines, and comparing against other PEFT methods such as Prompt Tuning or Visual Prompt Tuning.

**Strengths:**

1. Under a theoretical notion of expected adversarial robustness, this work investigate an overlooked but meaningful perspective, which is the relation between different fine tuning method and their adversarial robustness behavior.
2. The derived upper bound provides a useful insight, and the proposed LoRA initialization derived from it empirically improves adversarial robustness.
3. The concrete initialization scheme could be practically valuable for the PEFT community.

**Weaknesses:**

1. (Major) The assumption of bounded input is reasonable (line 155), but intermediate activations are neither assumed nor proved to be bounded.
   In particular, line 703 (Appendix A) and line 271 rely on this bounded-activation assumption when deriving the single-layer Transformer’s robustness bound and when generalizing to the multi-layer case, respectively.
   However, for deeper layers, if activations are not guaranteed to be bounded, the proof of Proposition 1 no longer holds for intermediate layers.
   It is therefore encouraged to clarify what precise assumption is used and to justify how intermediate activations could be considered bounded (or to add an explicit limitation discussion).

2. Adaptive defense strategies and other PEFT variants (e.g., Prompt Tuning, Visual Prompt Tuning) could be included for a more comprehensive comparison.

**Questions:**

As above

---

> ### Author Response · Authors · 2025-11-26
>
> We thank the reviewer for their constructive feedback, which has helped strengthen our manuscript. We particularly appreciate the recognition of our theoretical contributions and the helpful identification of areas requiring clarification. Below, we address the raised concerns and outline the revisions made.
>
> **W1 - Regarding the assumptions:** We apologize if this wasn’t clarified enough in our manuscript. In our problem setup (Section 3 - Line 153 to 155), we do indeed provide an assumption regarding the activation function, specifically that they are all assumed to be $1$-Lipschitz (which is actually the case of the majority of used functions, such as TanH …). As identified by the reviewer, such an assumption is important during the derivation of the upper bounds.
>
> **W2 - On the extension to other PEFT methods:** We thank the reviewer for pointing us in this direction. Indeed, generalizing the method to other PEFT strategies would be an interesting direction. In this perspective, and in line with the reviewer’s proposition, we extended the study to take into account prompt tuning, but also other variants of LoRA (such as DeLora and QLora) to showcase how our provided upper-bound can actually give some interesting insights about the robustness of these variants.

---

### Meta-Review · Area_Chair_12Qm · 2026-01-12

**Summary:**

This paper presents a theoretical and empirical analysis of the adversarial robustness of LoRA fine-tuning compared to head-only fine-tuning. The authors introduce a notion of expected adversarial robustness and derive bounds showing that, despite improved clean accuracy, LoRA can be inherently less robust due to the additional degrees of freedom in its low-rank adapters. They further analyze how the initialization of the low-rank matrices affects robustness and validate their claims through experiments on vision and language benchmarks under standard adversarial attacks. During the rebuttal, the authors addressed some of the reviewers’ concerns, but certain issues remain: (1) similar theoretical observations already exist in prior work; and (2) reduced adversarial robustness of LoRA compared to head-only fine-tuning is within expectation. These limits the contribution of this work to the research community. Therefore, we recommend rejecting this work.

**Reviewer Concerns:**

Most of the concerns have been addressed, but the following remain:

The study focuses on comparing the robustness of LoRA with head-only fine-tuning and theoretically and empirically verifies the conclusion that LoRA is less robust than head-only fine-tuning. However, this result is expected by Reviewer 8iVG and Reviewer zHsT and therefore is not particularly surprising.

The concern raised by Reviewer zHsT that similar theoretical results already exist in the literature on Lipschitz neural networks and Transformers has not been addressed.

**Reviewer Scores:**

Reviewer AbCa (score: 6, confidence: 1) would not change the score.

Other reviewers (all with score 4) would not change the score as the major concerns remain.

---

### Decision · Program_Chairs · 2026-01-26

Reject